# Grassland intensification effects cascade to alter multifunctionality of wetlands within metaecosystems

Yuxi Guo[1], Elizabeth H. Boughton[2] ✉, Stephanie Bohlman[3], Carl Bernacchi[4], Patrick J. Bohlen[5], Raoul Boughton[2], Evan DeLucia[6], John E. Fauth[5], Nuria Gomez-Casanovas[7,8], David G. Jenkins[5], Gene Lollis[2], Ryan S. Miller[9], Pedro F. Quintana-Ascencio[5], Grégory Sonnier[2], Jed Sparks[10], Hilary M. Swain[2] & Jiangxiao Qiu[1,11] ✉

Sustainable agricultural intensification could improve ecosystem service multifunctionality, yet empirical evidence remains tenuous, especially regarding consequences for spatially coupled ecosystems connected by flows across ecosystem boundaries (i.e., metaecosystems). Here we aim to understand the effects of land-use intensification on multiple ecosystem services of spatially connected grasslands and wetlands, where management practices were applied to grasslands but not directly imposed to wetlands. We synthesize long-term datasets encompassing 53 physical, chemical, and biological indicators, comprising >11,000 field measurements. Our results reveal that intensification promotes high-quality forage and livestock production in both grasslands and wetlands, but at the expense of water quality regulation, methane mitigation, non-native species invasion resistance, and biodiversity. Land-use intensification weakens relationships among ecosystem services. The effects on grasslands cascade to alter multifunctionality of embedded natural wetlands within the metaecosystems to a similar extent. These results highlight the importance of considering spatial flows of resources and organisms when studying land-use intensification effects on metaecosystems as well as when designing grassland and wetland management practices to improve landscape multifunctionality.

Securing food production while safeguarding natural capital remains one of the grand challenges in the 21st century and a top priority on the global policy agenda[1]. In an era of expanding population and wealth that leads to shifts towards resource-intensive diets, rising food demands are aggravating land-use conflicts and resource competition. While key to attaining food security and human livelihoods, agricultural intensification is an important driver of global change and significant contributor to rising environmental risks in the Anthropocene[2,3]. Climate change also poses substantial threats to resilience of agriculture, with disproportionate impacts on developing countries and marginalized communities[4,5]. It is thus vital to understand and explore transitions of agricultural systems towards a more 'sustainable intensification' paradigm[5,6] that encapsulates aims to boost productivity, improve ecosystem services, and bolster multifunctionality – the capacity of an ecosystem to simultaneously provide multiple functions or services.

As grasslands occupy ~25% of the Earth's land surface and 70% of agricultural production area[7], they are dominant terrestrial ecosystems and critical components of global food security. Grazing lands are responsible for 40% of agricultural output (e.g., meat and dairy

products) and support livelihoods of ~1.3 billion people worldwide[8]. Besides agricultural products, grasslands also deliver a wide array of essential ecosystem services[9]: they contribute to >10% of terrestrial net primary productivity, store up to 30% of global soil organic carbon[10–12], and serve as key habitats for diverse fauna and flora. At local to regional scales, grasslands provide valuable regulating services, such as soil health, flood abatement, nutrient retention, pollination and pest control[13–15].

Despite their social-ecological importance, grasslands remain understudied in sustainable development agendas[16]. This undermines the long-term capacity of grasslands to support biodiversity and provide society with goods and services needed to prosper. One key threat for grasslands is land-use intensification. Conceptually, grassland intensification is often perceived as a conversion of natural and semi-natural grasslands to intensively-managed or cultivated grasslands[17,18]. Yet it entails a portfolio of integrated practices such as introduction of non-native productive forage species, high-intensity livestock grazing, fertilization, drainage, and frequent mechanical soil disturbances[19]. Although intensification promotes forage and livestock production and contributes towards rural economic prosperity, it can also lead to persistent biodiversity loss and create unwanted declines and debts in other regulating and cultural ecosystem services[20,21], thus compromising grassland multifunctionality[22].

As boosting agricultural productivity is the major motivation of land-use intensification, tradeoffs from intensively-managed systems have been increasingly revealed[23–25]. A multifunctional approach is thus recognized as indispensable to consider the consequences of intensified management for a broad range of biodiversity, and ecosystem functions and services, but remains scarce[26–28]. Further, research addressing effects of land-use intensification on multiple ecosystem services and their interactions (i.e., tradeoffs and synergies) in grasslands has predominantly focused on temperate or semiarid regions[9]. Subtropical grasslands, with their unique and often humid climate and distinctive biophysical characteristics and management practices, have received far less attention but are facing acute threats from ongoing degradation[29]. Such knowledge is urgently needed, given that ~50% of the global population will reside in the subtropics and tropics by 2050[30], potentially leading to more intensified land uses and other anthropogenic modifications in this biogeographic region.

In addition, grasslands are not isolated, especially in the subtropical and tropical biomes, but rather integrate and interact with other ecosystems (e.g., wetlands or forests), forming complex 'metaecosystems', which are defined as a set of ecosystems connected by spatial flows of energy, materials, and organisms across ecosystem boundaries[31–33]. In the grassland-wetland metaecosystems, resource flows (e.g., driven by physical processes such as gravity or hydrological flows) and organismal movements can be significant spatial processes coupling ecosystems[31]. Examples include lateral nutrient and sediment transport from upland grasslands to embedded wetlands, and livestock and wildlife grazing and foraging in wetlands that moves nutrients to grasslands or dispersing plant species to other isolated wetlands. Some prior work (e.g., local empirical studies or global synthesis) has focused on land-use intensification effects on multifunctionality within grasslands[34,35]. However, limited research has explicitly addressed cascading effects of land-use intensification on multiple ecosystem services and their interactions within a metaecosystem. Such a comprehensive and cross-scale understanding from the lens of the metaecosystems is crucial for designing a multifunctional agricultural landscape and informing management decisions that often occur at larger spatial scales (e.g., watersheds) and can exert far-reaching impacts beyond focal production areas.

In this research, we aim to address these knowledge gaps by focusing on a comprehensively studied grassland-dominated landscape in Florida, USA (Fig. S1) that now experiences two levels of land-use intensity representative of the region – Semi-natural (SN, less intensively-managed and less altered from historic wet prairies) vs. Intensively-managed (IM, intensively-managed and completely converted from historic dry prairies)[36]. Compared to semi-natural grasslands, intensively-managed grasslands encompass integrated practices including fertilization, higher grazing pressure, conversion of native grasses into productive forages, and more intense drainage. Detailed definition, description, and comparison of two grassland types can be found in the Methods and Supplementary Information Table S1. Our study area is considered as an exemplar of humid subtropical grasslands in the U.S. and across the globe (e.g., those in Australia and South America)[37]. Subtropical grasslands are distinguished from their high latitude temperate counterparts by having dominant $C_4$ grass species and a warm and humid climate. In Florida, the wet-season humid climate and high groundwater table have created a landscape mosaic with numerous geographically isolated seasonal wetlands embedded in grasslands[36,38]. Geographically isolated wetlands are keystone landscape elements[39] providing many irreplaceable ecosystem services, particularly flood mitigation and nutrient retention, and are vital refugia for invertebrates, amphibians, and breeding and migratory birds. These wetlands are spatially coupled with grasslands and thus highly sensitive to surrounding grassland management, and have been substantially altered and lost due to historical and ongoing anthropogenic pressures[40].

We address four specific research questions: (1) How does land-use intensification affect a suite of grassland ecosystem services individually? (2) Do intensification effects cascade to natural wetlands embedded within grasslands? (3) What are consequences of land-use intensification for ecosystem service multifunctionality of spatially coupled grasslands and wetlands? and (4) How does land-use intensification alter interactions among multiple ecosystem services in grassland-wetland metaecosystems? To address these questions, we synthesized long-term datasets (2003–2020) of 53 different indicators (i.e., 29 for grasslands and 24 for wetlands) with >11,000 field measurements characterizing six major categories of ecosystem services (i.e., soil nutrient maintenance, water quality regulation, climate mitigation, biodiversity maintenance, invasion resistance, and agricultural production) (details in Tables 1, 2) and calculated various multifunctionality indexes. Indicators of these six ecosystem services were selected according to the Common International Classification of Ecosystem Services (CICES)[41,42] that encompass fundamental functions or properties responsible for service provision[21,43]. We transformed certain indicators when necessary, so that higher values always correspond to greater service provision. Specifically, we defined higher levels of soil nutrients, above- and below-ground primary production, biodiversity, forage nutrients and quantity, and cattle stocking density as desirable (from ecosystem service provision standpoint), because they represent greater supplies of soil nutrient maintenance, carbon storage, biodiversity maintenance, and agricultural production services, respectively[27,44]. We defined lower levels of water nutrients, greenhouse gas fluxes, and invasive species diversity as desirable, because they represent greater provision of water quality regulation, greenhouse mitigation, and invasion resistance services, respectively. We used long-term measurements so that our results are more robust, generalized, and less sensitive to temporal variations. We contrasted measurements between semi-natural (SN) and intensively-managed (IM) grasslands and their embedded natural wetlands (SN and IM wetlands hereafter, respectively) using standardized effect sizes to quantify and infer land-use intensification effects using linear mixed-effects models. Based on our analyses and results, we propose possible approaches to build sustainable intensification that fosters multifunctionality of metaecosystems.

**Table 1 | Summary of grassland ecosystem service indicators included in this study**

| Ecosystem service | Biophysical indicator | Temporal scale | Independent sample size | Independent observation | Measurement unit | Whether dataset has been previously published |
|---|---|---|---|---|---|---|
| Soil nutrient maintenance | Soil total nitrogen (TN) | 2016, 2019 | 133 | 63 | % | No |
| | Soil ammonium ($NH_4^+$) | 2019 | 72 | 72 | ug $g^{-1}$ | No |
| | Soil nitrate ($NO_3^-$) | 2016, 2019 | 141 | 71 | ug $g^{-1}$ | No |
| | Soil total phosphorus (TP) | 2016, 2019 | 144 | 72 | ug $g^{-1}$ | No |
| | Soil Mehlich-3 P | 2016, 2019 | 139 | 68 | lb $ac^{-1}$ | No |
| | Soil C/N ratio | 2016, 2019 | 144 | 72 | unitless | No |
| | Soil organic matter (OM) | 2016, 2019 | 142 | 71 | % | No |
| Water quality regulation | Water TN | 2008–2015 | 557 | 8 | mg $L^{-1}$ | No |
| | Water $NH_4^+$ | 2006–2015 | 714 | 8 | mg $L^{-1}$ | No |
| | Water $NO_3^-$ | 2008–2015 | 557 | 8 | mg $L^{-1}$ | No |
| | Water TP | 2003–2015 | 978 | 8 | mg $L^{-1}$ | No |
| | Water orthophosphate ($PO_4^{3-}$) | 2003–2015 | 976 | 8 | mg $L^{-1}$ | No |
| Carbon storage and greenhouse gas mitigation | Soil total carbon (TC) | 2016, 2019 | 135 | 63 | % | No |
| | Root biomass | 2016, 2019 | 144 | 72 | g | No |
| | Annual net primary productivity (ANPP) | 2017–2019 | 54 | 18 | g $m^{-2}$ | No |
| | $CO_2$ flux | 2013–2015 | 56 | 2 | g $m^{-2}$ $month^{-1}$ | Yes, Paudel et al., (2023); Gomez-Casanovas et al., (2018) |
| | $CH_4$ flux | 2013–2015 | 56 | 2 | g $m^{-2}$ $month^{-1}$ | Yes, Paudel et al., (2023); Gomez-Casanovas et al., (2018) |
| Biodiversity maintenance | Total plant richness | 2016–2019 | 48 | 8 | unitless | No |
| | Plant α-diversity | 2018–2019 | 32 | 8 | unitless | No |
| | Plant β-diversity | 2016–2019 | 48 | 8 | unitless | No |
| | Vertebrate richness | 2016–2018 | 44 | 44 | unitless | Yes, Tabak et al., (2019) |
| | Vertebrate α-diversity | 2016–2018 | 44 | 44 | unitless | Yes, Tabak et al., (2019) |
| Invasion resistance | Non-native plant richness | 2016–2019 | 48 | 8 | count | No |
| | Non-native vertebrate α-diversity | 2016–2018 | 44 | 44 | unitless | Yes, Tabak et al., (2019) |
| Agricultural production (forage and livestock) | Forage N | 2017–2019 | 286 | 8 | % | No |
| | Forage P | 2017–2019 | 284 | 8 | % | No |
| | Forage digestibility IVODM | 2017–2019 | 286 | 8 | % | No |
| | Palatable biomass cover | 2018–2019 | 96 | 24 | % | No |
| | Cattle stocking density | 2017–2018 | 192 | 8 | d $ac^{-1}$ $month^{-1}$ | No |

## Results

### Effects on grasslands

Intensified management in upland grasslands led to significant differences in at least half of the indicators for each ecosystem service category when compared to semi-natural management (Fig. 1). First, intensively-managed grasslands showed higher soil nutrients than semi-natural grasslands, as reflected by almost all indicators, including ammonia ($NH_4^+$), total phosphorus (P), Mehlich-3 P (i.e., plant-available P), organic matter (OM) content, and carbon-to-nitrogen (C/N) ratio (Fig. 1A). Consequently, ditch waterbodies adjacent to intensively-managed grasslands had higher P levels (i.e., as shown by higher orthophosphate ($PO_4^{3-}$) and total P concentrations), albeit lower total N, as compared to those in adjoining semi-natural grasslands (Fig. 1B). In addition, greater root biomass and aboveground primary productivity along with more methane (i.e., $CH_4$) emissions were found in intensively-managed than semi-natural grasslands (Fig. 1C). Biodiversity metrics were lower in intensively-managed than semi-natural grasslands, as shown by significantly lower vascular plant α- and β-diversity and moderately lower vertebrate diversities (Fig. 1D). Further, invasion resistance was higher in semi-natural than intensively-managed grasslands, which had greater

non-native plant richness (Fig. 1E). Our results were consistent even if we only analyzed non-native plant richness for those that are non-planted (Fig. S2). As for provisioning services, intensified management supported higher quality and quantity of forage production, where intensively-managed grasslands produced forages with significantly higher P content and in vitro organic matter digestibility (IVOMD), moderately higher palatable forage coverage, and served higher cattle stocking density as compared to semi-natural grasslands (Fig. 1F).

### Effects on embedded wetlands

Land-use intensification in upland grasslands exerted cascading effects to embedded natural wetlands, where most of the management practices were not directly implemented. Within each ecosystem service category, differences in at least one indicator were detected between semi-natural and intensively-managed wetlands. Among wetland soil nutrients, only total soil P was greater in intensively-managed than semi-natural wetlands (Fig. 2A). However, intensively-managed wetlands showed lower water quality than semi-natural wetlands, as evidenced by greater total N, total P, and $PO_4^{3-}$ concentrations (Fig. 2B). Similar to surrounding grasslands, wetlands also

**Table 2 | Summary of wetland ecosystem service indicators included in this study**

| Ecosystem service | Biophysical indicator | Temporal scale | Independent sample size | Independent observation | Measurement unit | Whether dataset has been previously published |
|---|---|---|---|---|---|---|
| Soil nutrient maintenance | Soil total nitrogen (TN) | 2007, 2009, 2016 | 60 | 20 | % | Yes, Ho et al., (2018) |
| | Soil total phosphorus (TP) | 2007, 2009, 2016 | 60 | 20 | % | Yes, Ho et al., (2018) |
| | Soil C/N ratio | 2007, 2009, 2016 | 60 | 20 | unitless | Yes, Ho et al., (2018) |
| | Soil organic matter (OM) | 2007, 2009, 2016 | 60 | 20 | % | Yes, Ho et al., (2018) |
| Water quality regulation | Water TN | 2006, 2008, 2009, 2014 | 80 | 20 | mg L$^{-1}$ | Yes, Jansen et al., (2019) |
| | Water NH$_4^+$ | 2006, 2008, 2009, 2014 | 80 | 20 | mg L$^{-1}$ | Yes, Jansen et al., (2019) |
| | Water NO$_3^-$ | 2008, 2009, 2014 | 60 | 20 | mg L$^{-1}$ | No |
| | Water TP | 2006, 2008, 2009, 2014 | 80 | 20 | mg L$^{-1}$ | Yes, Jansen et al., (2019) |
| | Water PO$_4^{3-}$ | 2006, 2008, 2009, 2014 | 80 | 20 | mg L$^{-1}$ | Yes, Jansen et al., (2019) |
| Carbon storage and greenhouse gas mitigation | Soil total carbon (TC) | 2007, 2009, 2016 | 60 | 20 | % | Yes, Ho et al., (2018) |
| | Root biomass | 2015 | 25 | 4 | g | Yes, DeLucia et al., (2019) |
| | Annual net primary productivity (ANPP) | 2016, 2017 | 40 | 20 | g m$^{-2}$ | Partial data published in Sonnier et al., (2020) |
| | CH$_4$ flux | 2013–2015 | 188 | 16 | umol m$^{-2}$ s$^{-1}$ | Yes, DeLucia et al., (2019) |
| Biodiversity maintenance | Total plant species richness | 2006–2020 | 240 | 20 | count | No |
| | Plant species α-diversity | 2006–2020 | 240 | 20 | unitless | No |
| | Invertebrate richness | 2006 | 20 | 20 | count | Yes, Medley et al., (2015) |
| | Invertebrate α-diversity | 2006 | 20 | 20 | unitless | Yes, Medley et al., (2015) |
| | Vertebrate richness | 2006, 2008, 2009 | 60 | 20 | count | Yes, Medley et al., (2015) |
| | Vertebrate α-diversity | 2006, 2008, 2009 | 60 | 20 | unitless | Yes, Medley et al., (2015) |
| Invasion resistance | Non-native plant richness | 2006–2020 | 240 | 20 | count | No |
| Agricultural production (forage) | Forage N | 2006–2008 | 404 | ~135 | % | Yes, Sonnier et al., (2020) |
| | Forage P | 2006–2008 | 380 | ~135 | % | Yes, Sonnier et al., (2020) |
| | Forage C/N ratio | 2006–2008 | 404 | ~135 | unitless | Yes, Sonnier et al., (2020) |
| | Palatable biomass cover | 2018, 2020 | 40 | 20 | % | Yes, Sonnier et al., (2020) |

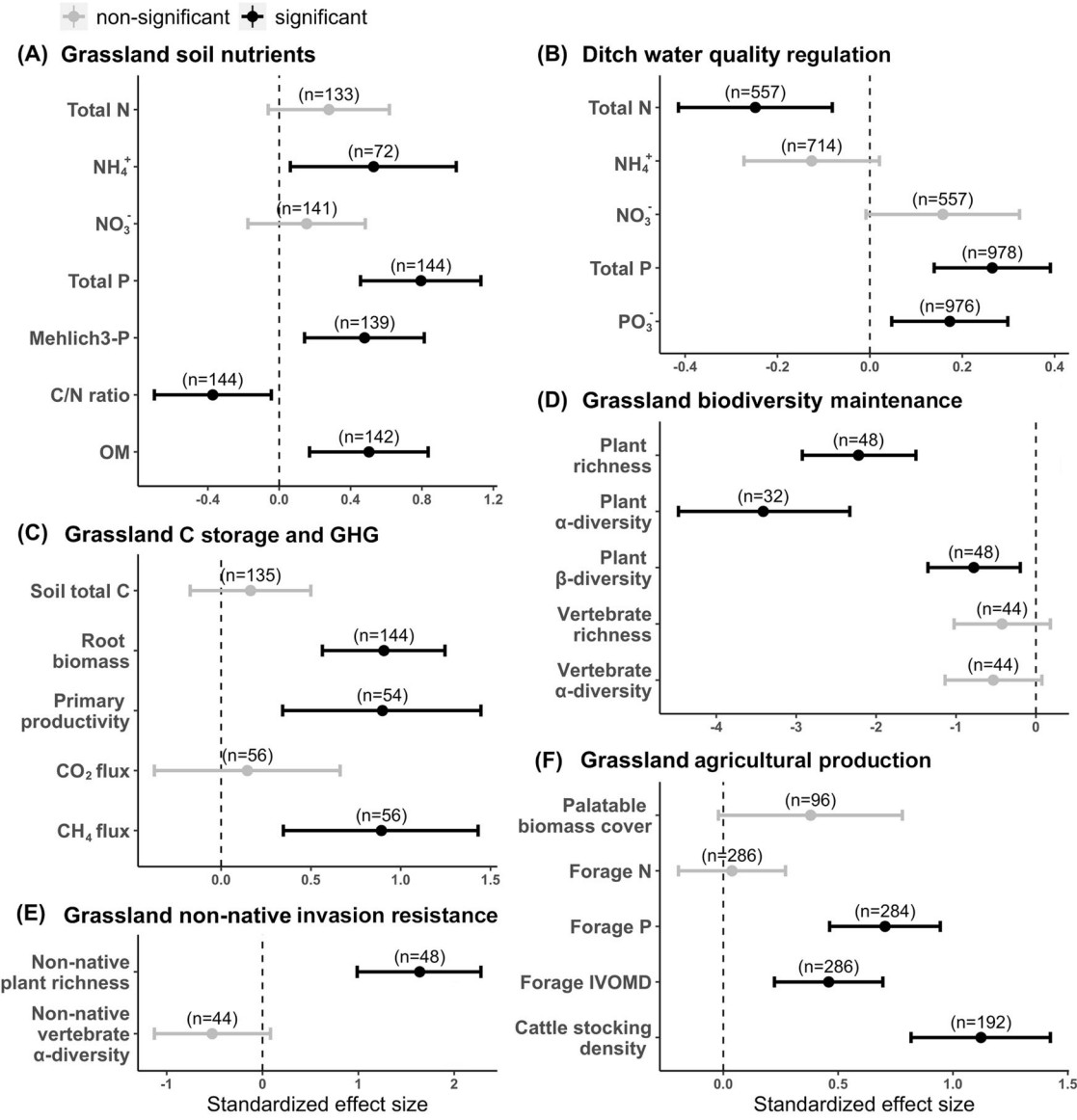

**Fig. 1 | Standardized effect sizes (Hedge's *d*; center for the error bars) of land-use intensification on indicators of multiple ecosystem services in grasslands.** **A** Soil nutrient maintenance; (**B**) Adjacent ditch water quality regulation; (**C**) Carbon storage and greenhouse gas (GHG) mitigation; (**D**) Biodiversity maintenance; (**E**) Invasion resistance; and (**F**) Agricultural production. Effect sizes of intensification were estimated by comparing Intensively-managed (IM) vs. Semi-natural (SN) grasslands, with error bars representing 95% confidence intervals. Positive Hedge's d denotes a higher indicator value for IM than SN grasslands. Black bars represent significant differences ($\alpha \leq 0.05$) between IM and SN grasslands, whereas grey bars indicate non-significant differences. Numbers in parentheses mean the sample size for estimating the effect size of each indicator. Source data are provided as a Source Data file.

showed mixed responses on carbon storage and greenhouse gas fluxes. Compared to semi-natural wetlands, intensively-managed wetlands had lower root biomass, but higher aboveground primary productivity and greater soil-level $CH_4$ emissions (Fig. 2C). Land-use intensification in grasslands resulted in lower biodiversity metrics in embedded wetlands, as indicated by lower plant and ectothermic vertebrate diversities in intensively-managed than semi-natural wetlands (Fig. 2D). As compared to semi-natural wetlands, intensively-managed wetlands provided forage with higher N and P contents, but had less palatable forage coverage (Fig. 2E) and more non-native plant species (Fig. 2F).

### Effects on ecosystem service multifunctionality

Responses of spatially coupled grasslands and wetlands to two levels of land-use intensities were assessed using several multifunctionality (MF) indexes by including all ecosystem service indicators (i.e., 29 for

grasslands and 24 for wetlands). MFs were calculated and compared across four commonly adopted approaches, including simple averaging, quantile-based threshold, service-based weighted averaging, and cluster-based weighted averaging MFs[22,45,46]. Our results demonstrated that MFs were 9.7–27.3% lower (mean = 15.5%) in intensively-managed than semi-natural grasslands. MFs were 6.9% – 23.9% lower (mean = 13.0%) for intensively-managed than semi-natural wetlands (Fig. 3). Although absolute MFs calculated from the four different approaches induced slight variations, the overall direction and trend of land-use intensification effects on MFs were consistent (Fig. 3, Table S2). The relative contribution of individual ecosystem service categories to overall multifunctionality was illustrated by the averaged standardized effect size of indicators under each service (Fig. 4). Intensively-managed grassland-wetland metaecosystems resulted in greater soil fertility and agricultural production, but lower water quality, reduced plant and vertebrate diversity,

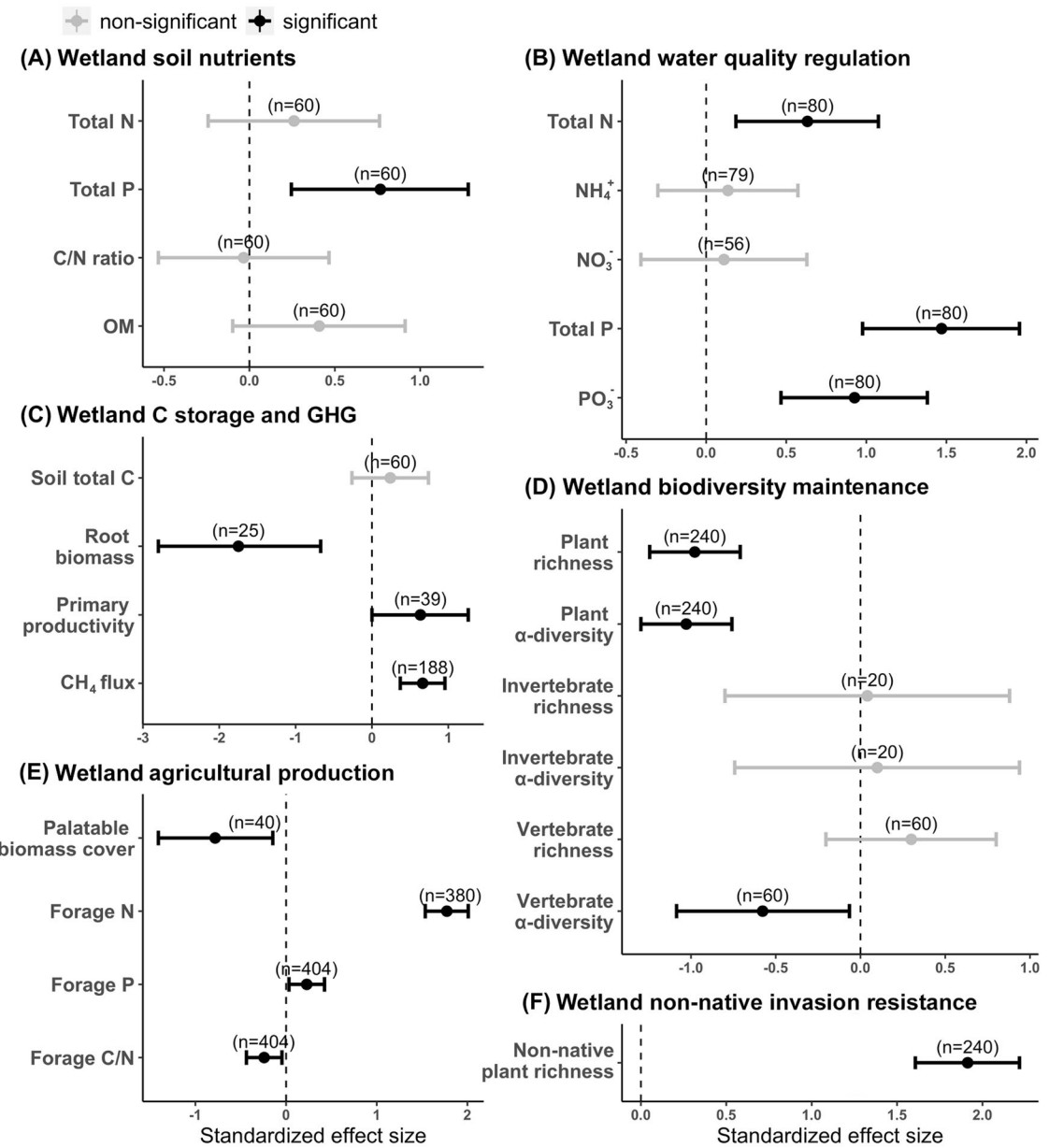

**Fig. 2 | Standardized effect sizes (Hedge's *d*; center for the error bars) of land-use intensification on indicators of multiple ecosystem services in embedded wetlands. A** Soil nutrient maintenance; (**B**) Water quality regulation; (**C**) Carbon storage and greenhouse gas mitigation; (**D**) Biodiversity maintenance; (**E**) Agricultural production; and (**F**) Invasion resistance. Effect sizes were estimated by comparing wetlands embedded in Intensively-managed (IM) vs. Semi-natural (SN) grasslands, with error bars representing 95% confidence intervals. Positive Hedge's d denotes a higher indicator value for IM than SN wetlands. Black bars represent significant differences ($\alpha \leq 0.05$) between IM and SN wetlands, whereas grey bars indicate non-significant differences. Numbers in parentheses mean the sample size for estimating the effect size of each indicator. Source data are provided as a Source Data file.

and more invasive plant species, as compared to their semi-natural counterparts (Fig. 4).

**Effects on ecosystem service relationships**

Significant interactions existed among ecosystem service indicators, some of which were altered by land-use intensification. In grasslands, there were positive correlations between soil nutrients and forage quality (Fig. 5A, B), and between cattle stocking density and plant β-diversity (Fig. 5C), which were consistent across land-use intensities. However, plant α-diversity and forage N content were negatively correlated in intensively-managed grasslands, but were uncorrelated in semi-natural grasslands (Fig. 5D). Similarly, positive correlations between soil total C and root biomass (Fig. 5E), and negative correlations between soil nutrient (C/N ratio) and non-native plant diversity

were found in semi-natural but not in intensively-managed grasslands (Fig. 5F).

In embedded natural wetlands, certain paired ecosystem service relationships were unaffected by land-use intensification, such as positive correlations between water total P and forage P (Fig. 6A), and negative correlations between water total P and plant diversity (Fig. 6B) and between water NH$_4^+$ and ectothermic vertebrate diversity (Fig. 6C). However, upland intensification altered some wetland eco-system service relationships. For example, positive correlations between soil total P and wetland primary productivity (Fig. 6D) and negative correlations between soil OM and plant richness (Fig. 6E) only existed in semi-natural wetlands, whereas negative correlations between soil total P and water total P (Fig. 6F) were significant only in intensively-managed wetlands.

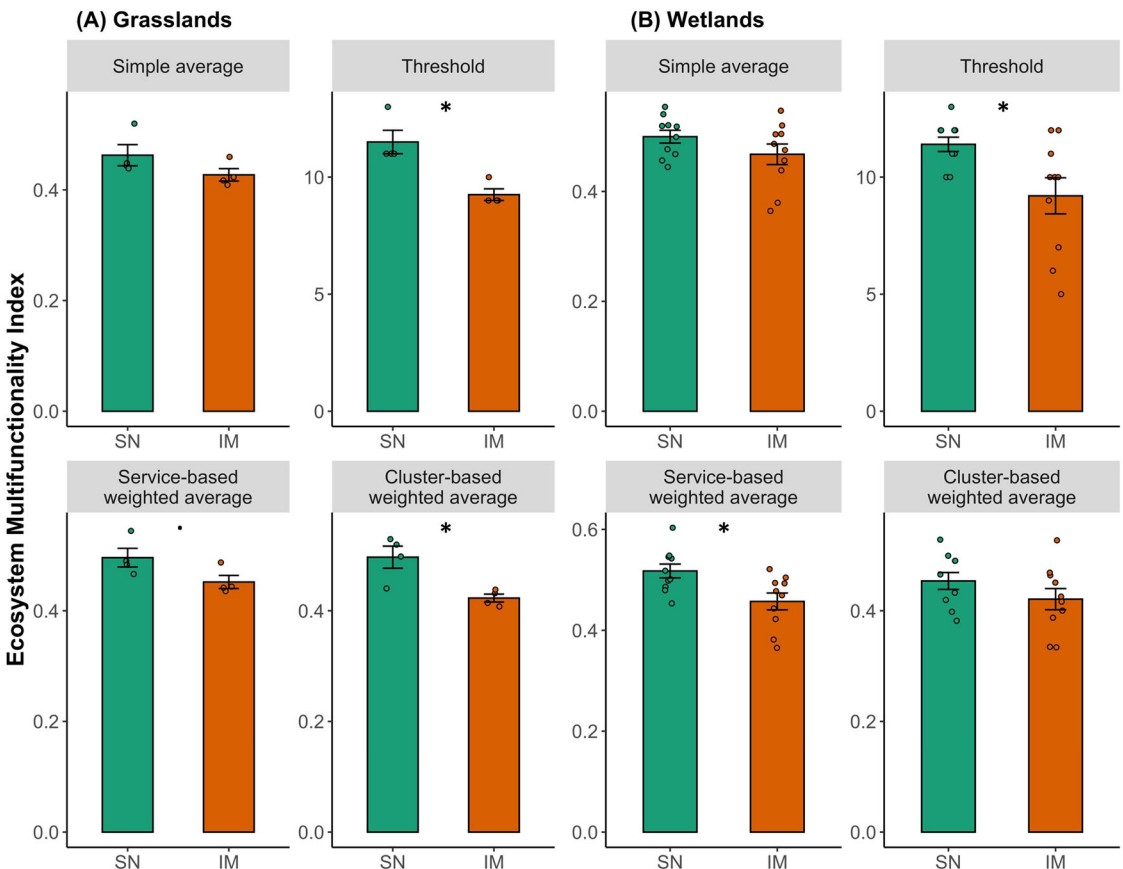

**Fig. 3 | Effects of agricultural land-use intensification on ecosystem service multifunctionality.** Multifunctionality of (**A**) grasslands ($N = 8$) and (**B**) embedded natural wetlands ($N = 20$) was quantified using four different approaches: (1) Simple average of all indicators; (2) Top 50% quantile-based threshold; (3) Service category-based weighted average; and (4) Cluster-based weighted average. Data were presented as mean values ± standard errors (SEs). Level of statistical significance: •$p < 0.1$; *$p < 0.5$ from the Kruskal–Wallis test. Exact $p$ values were provided in the Supplementary Table S2. IM – Intensively-managed (orange); SN – Semi-natural ecosystems (green). Source data are provided as a Source Data file.

## Discussion

Our study integrates multifunctional approaches and the metaecosystems framework to investigate land-use intensification effects on spatially connected ecosystems. Our results demonstrate that land-use intensification profoundly altered a broad suite of ecosystem functions and services and their relationships in grasslands, with cascading impacts on embedded natural wetlands where most of the management practices were not directly imposed (with the exception of grazing; details in Methods). Our results highlight the importance of understanding land-use intensification effects on multifunctionality of metaecosystems through considering spatial flows of resources and organisms across coupled ecosystems, and suggests the need for better quantification and investigation of these spatial flows.

### Land-use intensification effects on grassland-wetland metaecosystems

Land-use intensification increased soil nutrients, improved forage production and quality, and lower P-related water quality. These effects could be associated with historical P fertilization, ongoing N fertilization, periodic liming, and conversion of natural vegetation to forage grasses. First, fertilization and liming are key to intensively-managed grasslands dominated by bahiagrass (*Paspalum notatum* Flüggé), a predominant forage species in southeastern U.S. and other tropical and subtropical biomes. Bahiagrass adapts well to acidic sandy soils and is highly resistant to diseases and pests[47]. Typically, bahiagrass has lower nutritional values than normal $C_3$ grasses in infertile soils[48], making it inadequate for livestock[48]. However, when fertilizers are applied, bahiagrass can improve its nutrient uptake and retention capacities than other common grasses in Florida[49], resulting in improved nutritional quality. Hence, N fertilization is vital in ranching operations to improve nutritional quality of bahiagrass, which further leads to increased soil nutrients. In addition, legacy soil P from previous applications and continued N fertilization along with extensive drainage under intensified practices also likely contributed to lowered water quality in nearby ditches and adjoining wetlands[50].

Replacement of native plant species with introduced forage grasses was associated with reduced invasion resistance and plant α- and β-diversity in intensively-managed grasslands and wetlands. Lower invasion resistance (quantified using non-native plant richness) was found in intensively-managed than semi-natural grasslands, regardless of whether intentionally-planted non-native plants were included in this analysis or not (Fig. S2), indicating robustness of land-use intensification effects on invasion resistance. Besides conversion to bahiagrass, other integrated practices under IM approaches, including fertilization, drainage, and intensive grazing could also contribute to decreased plant diversity. Specifically, nutrient runoff (either from local fertilization in grasslands or their lateral flows to wetlands) has been showed to reduce native plant diversity by favoring fast-growing species[51,52]. Heavy grazing inherent in intensively-managed systems can further exacerbate this plant diversity reduction through selection of grazing-resistant species[53]. Indeed, prior work in our study system has shown that the combination of higher nutrient runoff and higher stocking rate likely explained lowered plant diversity in IM wetlands dominated by a few grazing avoiders (e.g., *Juncus effusus* subsp.

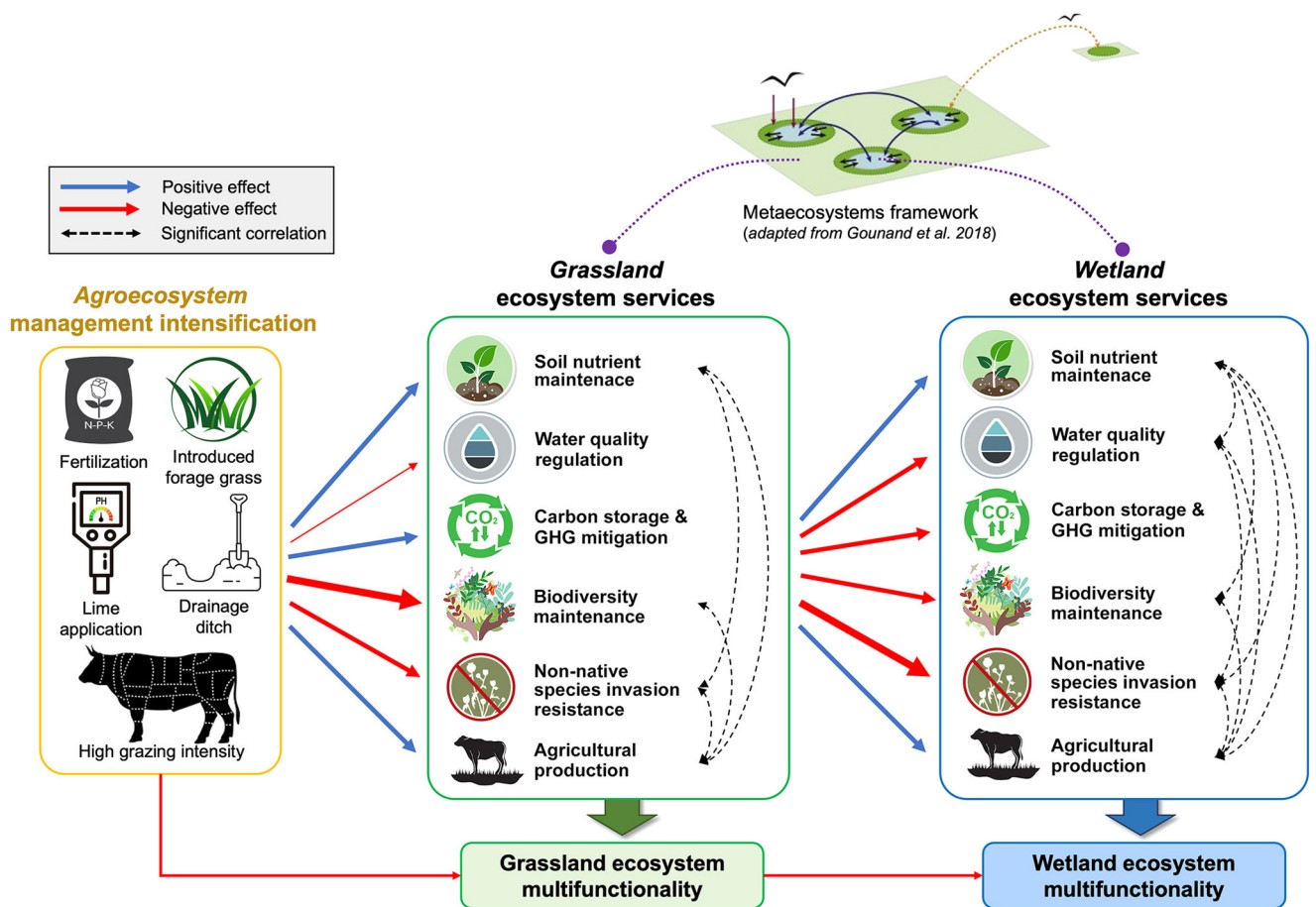

**Fig. 4 | Conceptual diagram illustrating how multiple ecosystem services from grassland-wetland metaecosystems were affected by upland land-use intensification.** Blue arrows indicate positive effects and red arrows denote negative effects (from the perspective of ecosystem service provision) due to intensification. Width of the arrow corresponds to the magnitude of effect size, calculated by averaging the standardized effect size of all indicators within each ecosystem service category. Dotted arrows indicate significant relationships in biophysical indicators across ecosystem service categories.

*solutus*)[37,54,55]. Moreover, excessive nutrients in IM wetlands could also contribute to reduced native plant diversity by enhancing algal growth and light competition[56,57], thus favoring tall perennial and unpalatable macrophytes and simplifying plant community composition. Such effects could cascade to affect other taxa and trophic groups[58]. This is evidenced by decreased ectothermic vertebrates diversity in IM wetlands, presumably due to uniform vegetation structure and homogenous habitats in these wetlands[59,60].

Greenhouse gas mitigation service in grassland-wetland metaecosystems was also susceptible to land-use intensification (Figs. 1C, 2C). Specifically, greater $CH_4$ emissions could be associated with multiple intensified practices acting in concert, including higher grazing density, N fertilization, and hydrological modification. Prior research at our study region indeed demonstrated that increased soil wetness due to higher stocking intensity and extensive drainage, along with fertilizer N and N input from urine could increase $CH_4$ emissions[61–63]. Greater enteric fermentation and manure deposition associated with higher cattle stocking density under intensified practices could also lead to greater $CH_4$ emissions in intensively-managed than semi-natural systems[64].

### Considerations for sustainable land management

Our findings point to three important considerations for sustainable land management: (1) a multifunctionality and landscape perspective for sustainable intensification; (2) the metaecosystems framework to assess land-use intensification and its spatially cascading effects; and (3) examination of intensification-induced variations in tradeoffs and synergies across scales.

First, agricultural intensification often fails to achieve simultaneous positive ecosystem service and wellbeing outcomes[65], suggesting a need for alternative pathways to sustainable agriculture. Our results revealed that the two focal management intensities produced complementary outcomes in the supply of multiple services (Figs. 3, 5), where IM (relative to SN) led to improved agricultural productivity at the expense of water quality degradation, biodiversity loss, more $CH_4$ emissions and non-native species, and thus net loss in multifunctionality. In addition to these differences in ecosystem services, a recent economic model[66] (with data collected at the ranch level) using calf prices from 2012 to 2020 and estimates of beef production based on industry standard stocking and weaning rates, further showed that intensively-managed grasslands generated $852.5 per hectare from calf production compared to $292.3 per hectare in semi-natural grasslands. Thus, intensively-managed grasslands provided up to three times as much economic gain as semi-natural grasslands per area unit[66]. In other words, if only semi-natural approaches were practiced, production areas would have to be expanded in area by three times to achieve equivalent economic returns, or similarly by three times if calculated on a stocking density basis (i.e., $36.8 \pm 1.2$ animal use days (AUD) ha$^{-1}$ month$^{-1}$ in IM vs. $14.3 \pm 0.7$ AUD ha$^{-1}$ month$^{-1}$ in SN grasslands). Hence, a shift from the current ~50:50 IM:SN ratio to a higher proportion of SN is not presently economically sustainable, and thus cannot support a viable grazing-based ranch economy. Economically viable ranches are critical to rural community wellbeing, as currently revealed and addressed in the U.S. Department of Agriculture Long-term Agroecosystem Research (LTAR) network[67]. On the other hand, it

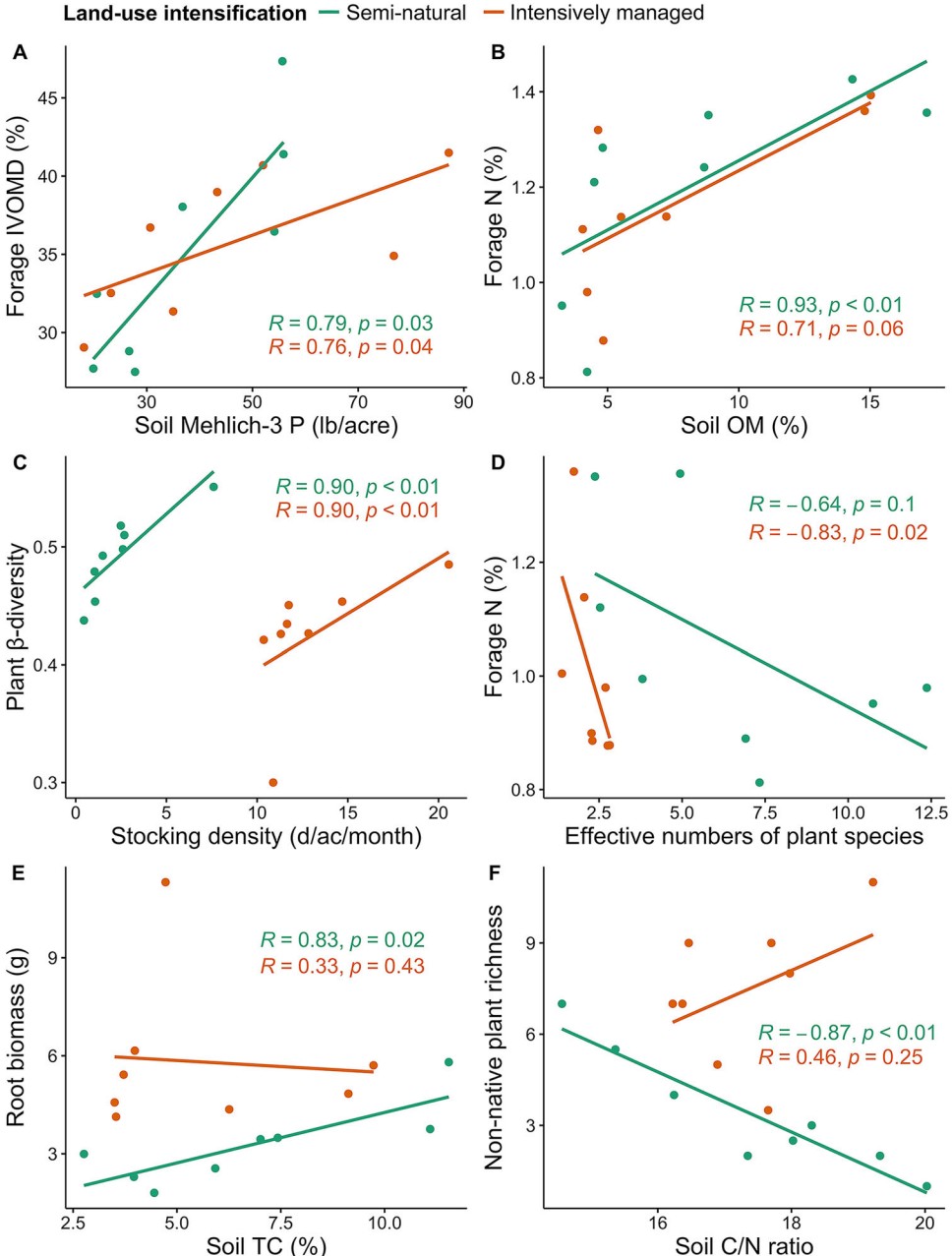

**Fig. 5 | Spearman's rank correlations between ecosystem service indicators in Intensively-managed (orange) and Semi-natural (green) grasslands. A** Soil Mehlich-3 phosphorus (P) vs. forage in-vitro organic matter digestibility (IVOMD); (**B**) soil organic matter (OM) vs. forage nitrogen (N); (**C**) cattle stocking density vs. plant beta (β) diversity; (**D**) effective numbers of plant species (calculated based on plant α-diversity) vs. forage N; (**E**) soil total carbon (TC) vs. root biomass; and (**F**) soil carbon to nitrogen (C/N) ratio vs. non-native plant richness. Exact p values of presented correlations were shown in the figures. Source data are provided as a Source Data file.

would not make sense to convert SN to IM, because SN provides forage stability and diversified ranch revenues (e.g., hunting)[68]. Even if it makes economic sense to do so, converting IM to SN would likely be highly difficult and expensive, because IM appears to be a stable de novo grassland state that has resilient non-native grasses and confers a long-lasting soil P legacy. As a result, reducing management intensity alone will not revert intensively-managed to semi-natural or native grasslands, nor increase ecosystem service multifunctionality, without substantial restoration efforts. Hence, neither intensively-managed nor semi-natural systems alone would qualify as sustainable intensification. Rather, sustainable intensification is more likely attained at the landscape scale where both management intensities are included and spatially distributed in a mosaic[21]. Such landscape-level strategies can

be further optimized by spatially targeting local management practices (i.e., incorporating the 'land sparing' concept[69]) to obtain complementarity and a good compromise among agricultural production, biodiversity conservation, ecosystem services, and financial returns.

Second, the metaecosystems offer an ideal theoretical and practical framework to evaluate land-use intensification beyond production areas that considers spatially cascading effects across coupled ecosystems[14,31,33,70]. IM in our study system led to declined multifunctionality of embedded wetlands, where most agricultural management practices were not directly imposed (Figs. 3, 4). Thus, it is important to account for spatially displaced negative impacts resulting from local intensification, which are often neglected, less well quantified, and seldom considered in agricultural management and decision-

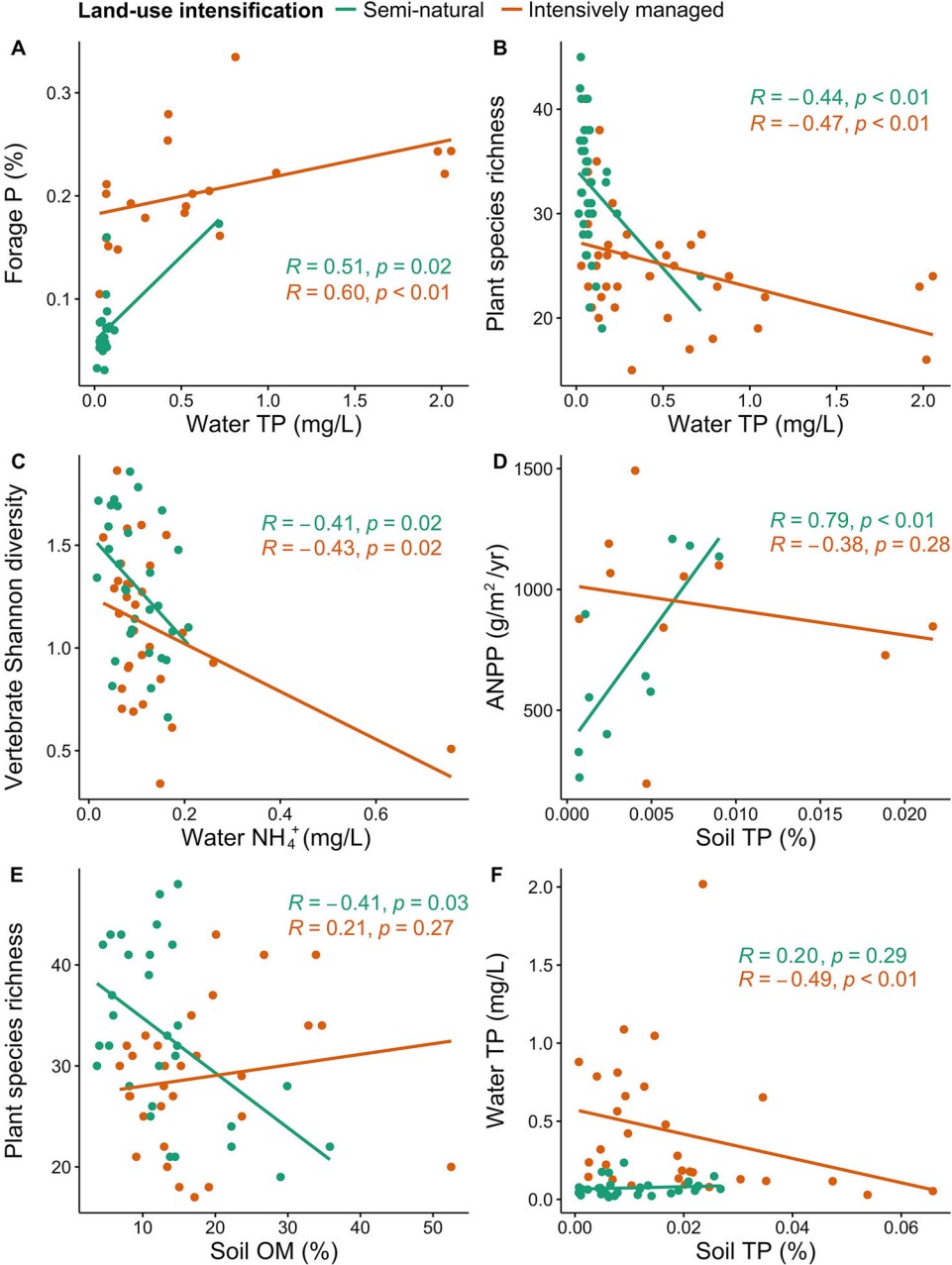

**Fig. 6 | Spearman's rank correlations between ecosystem service indicators in Intensively-managed (orange) and Semi-natural (green) wetlands. A** Water total phosphorus (P) vs. forage P; (**B**) water total P (TP) vs. plant species richness; (**C**) water ammonia ($NH_4^+$) concentration vs. ectothermic vertebrates Shannon diversity; (**D**) soil total phosphorus (TP) vs. aboveground net primary productivity (ANPP); (**E**) soil organic matter (OM) vs. plant species richness; and (**F**) soil Total P vs. water Total P. Exact $p$ values of presented correlations were shown in the figures. Source data are provided as a Source Data file.

making. The same approach applies at vast spatial scales, such as intensive agriculture and prairie potholes in the Midwest[43], and high-intensity irrigated agriculture in Australia's Murray–Darling Basin. Adopting the metaecosystems framework also implies opportunities for interventions. For example, some grassland management such as patch-burn grazing can improve nutrient-use efficiency and forage quality[66,71], which may reduce reliance on costly fertilizer inputs and ameliorate nutrient loads flowing to embedded wetlands. Payment for ecosystem services could also shift incentives toward hydrological and nutrient retention in the connected wetlands, and foster best management practices (e.g., buffer strips) to offset regional effects of grassland intensification[72].

Finally, land-use intensification altered tradeoffs or synergies among ecosystem services, consistent with other studies[17,73,74]. Such

effects can occur not only where intensification takes place, but also in other spatially coupled ecosystems. For example, land-use intensification decoupled synergies between soil nutrients and productivity in grasslands and wetlands, due to nutrient saturation[75,76]. Emerging tradeoffs between plant diversity and forage quality were detected in intensively-managed grasslands, due to dominant bahiagrass having higher forage N than mixtures of plant species[77]. Further, upland intensification decoupled tradeoffs between soil nutrients and plant species richness in wetlands, though not in grasslands. Such different responses between grasslands and wetlands were likely due to nutrient retention in wetlands when nutrients applied on grazed upland grasslands moved into embedded wetlands. Thus, wetlands were more affected than grasslands in this soil nutrient – plant diversity relationship[78]. Interestingly, across all possible paired services, land-

use intensification weakened the magnitude of most ecosystem service relationships. Weakened linkages among functions or services might induce uncertainties in earth system models for predicting environmental change effects on ecosystem services in agriculture-dominated metaecosystems[79]. Hence, it is crucial to consider such dynamic shifts in ecosystem service relationships if sustainable intensification aims to take advantage of synergies and mitigate unwanted tradeoffs.

## Study limitations and future research needs

Our research has several limitations worth noting that indicate future research needs. (1) This study focuses on the most dominant and commonly adopted semi-natural and intensively-managed practices in subtropical grassland ecosystems in the U.S. Our study, to the best of our knowledge, includes the most comprehensive dataset available for subtropical grasslands in the U.S. and perhaps globally, and identifies new avenues of investigation into multifactor interactions within metaecosystems. The depth of dataset and analyses offer a unique opportunity for future comparative research with similar low-latitude ecosystems (e.g., tropics of Australia, Pantanal in South America) and agricultural-based metaecosystems where scale, local contexts, and farming practices could vary and modify generalizability of our findings. This analysis also alludes to intricate mechanisms underpinning integrated intensification practices in managed grasslands, setting the stage for further enriching the body of knowledge in this field to disentangle relative importance of different mechanisms. (2) Our analyses center on land-use intensification as the key driver for grasslands. Continued long-term measurements (e.g., leveraging LTAR or other long-term research programs) using the metaecosystems framework outlined here may help to resolve potential consequences from other natural and anthropogenic changes, such as climate change and altered disturbance regimes, and how they interact with land-use intensification to alter metaecosystems multifunctionality. (3) The metaecosystems framework, by definition, embraces spatial flows across ecosystem boundaries. Despite measuring 53 indicators spanning nearly two decades, spatial flows were only inferred (not directly quantified) from spatial associations of grasslands – wetlands according to hypothesized cascading effects and observed changes in ecosystem services. Major new efforts are required to explicitly measure dominant spatial flows (e.g., resources, organisms), while examining interactions among multiple spatial flows and scales at which such flows are imperative to drive metaecosystems multifunctionality. Further, land-use intensification effects on multifunctionality in our metaecosystems might be dominated by resource flows from grasslands to wetlands, owing to physical geography and how these processes might operate laterally across the landscape. Nevertheless, biotic processes and flows can also occur reciprocally from wetlands to grasslands (e.g., subsidies of aquatic life that supports upland food webs, grazers' nutrient transport to uplands from wetlands), but have not been measured and analyzed in this work. These are fruitful research avenues to empirically support the metaecosystems theory development and provide mechanistic understanding of dominant spatial processes that underlie land-use intensification effects on metaecosystems.

## Concluding remarks

Our research reveals direct and cascading effects of land-use intensification on ecosystem service multifunctionality of spatially coupled grasslands and wetlands, and demonstrates the importance of landscape-level strategies to achieve sustainable agriculture intensification. Specifically, we found that: (1) intensification promoted provisioning services including forage and livestock production, but at the expense of regulating and supporting services, including water quality regulation, greenhouse gas mitigation, biodiversity maintenance, and non-native species invasion resistance (Fig. 4); (2) consistent cascading effects from grassland intensification were observed in spatially connected wetlands (Fig. 4); and (3) intensification altered and tended to weaken ecosystem service relationships in both grasslands and wetlands. Our results highlight that neither of the two land management intensities (i.e., semi-natural and intensively-managed) performs better than the other, but rather they are complementary in their supported functions and services. Hence, these two management intensities should continue to be implemented together and spatially optimized at the landscape scale to achieve sustainable intensification.

Our findings contribute to the growing literature on ecosystem service multifunctionality[22,80] and sustainable intensification, but highlight the need to incorporate the metaecosystems framework. This study fills a key empirical data and knowledge gap on responses of multiple ecosystem services to agricultural intensification in humid tropical and subtropical regions, which could help quantify and predict anthropogenic effects on ecosystem service multifunctionality at regional and global scales. Results can also inform the development of agroecosystem sustainability indicators and metrics that encompass production, environment, and social domains[81], which are being adopted across a range of grazing land sites (e.g., LTAR in the U.S.) and globe. Our research leads to insights into potential displaced ecological costs for economic gains in grassland-wetland metaecosystems, which could be generalizable for similar hydrologically-mediated metaecosystems. Future research that links our ecosystem service multifunctionality results with socio-economic data to provide insights into multifunctional, profitable, resilient, and equitable agricultural landscapes could guide sustainable intensification of agroecosystems in an era of the Anthropocene with dwindling natural resources and rapid environmental changes.

## Methods
### Study region

Our study area is a working ranch (Archbold Biological Station's Buck Island Ranch; BIR) in south-central Florida, USA (27°09′N, 81°11 W) (Fig. S1) that has been historically managed at two intensity levels (Table S1) with full commercial operations (i.e., 4,252 ha) for research and educational purposes[36]. The central and north-central areas of BIR are classified as Intensively-managed (IM) grasslands, after being heavily drained and converted to more productive forage grasses (e.g., *Paspalum notatum* Flüggé) in the 1960s (Fig. S1). Intensively-managed grasslands also received nitrogen (N), phosphorus (P), and potassium (K) fertilizers with regular lime application. However, P and K fertilizer applications, historically applied using $P_2O_5$ and $K_2O$ at a rate of 34 – 90 kg ha$^{-1}$, were ceased in 1986[37]. N fertilization is still being applied using $NH_4SO_4$ or $NH_4NO_3$ at a rate of 56 kg ha$^{-1}$ annually or semi-annually[37]. All fertilizers applied followed the best management practices guidelines at the time of application. Grazing has been practiced in intensively-managed grasslands since the 1970s, with more intense activities in the wet season and a typical cattle density of 0.57–1.5 cows per hectare. The prairies and savannas surrounding IM grasslands in BIR are classified as Semi-natural (SN) (i.e., low land-use intensity) grasslands (Fig. S1), which were less drained, only partially converted to *P. notatum*, never fertilized, and did not have a history of other soil amendments. Semi-natural grasslands have been moderately grazed, predominantly during the dry season with a typical cattle density of 0.15–1.12 cows per hectare.

BIR has a humid subtropical climate with a distinct hot wet season (June–October) and a cool dry season (November-May). Average daily temperatures during the two seasons are 26.1 °C and 19.4 °C, respectively. Average annual precipitation is ~1300 mm, of which 70% occurs during the wet season. More than 600 seasonal and isolated wetlands are interspersed across the landscapes (accounting for 15% of BIR area) (Fig. S1) and spatially coupled with grasslands, forming metaecosystems. While different intensities of land management are directly imposed upon grasslands, their effects can presumably cascade to

alter embedded wetlands through spatial flows at landscape scales (e.g., resource flow, foraging, dispersal, life-cycle migration). One exception is grazing, since natural wetlands embedded within grasslands are not fenced and can potentially be subject to light grazing activities from cattle (e.g., for cooling needs and as additional food sources). As a part of the LTAR network since 2014, BIR has been conducting long-term ecological monitoring, assessment, experiments, and field measurements (e.g., soil nutrients, water quality, greenhouse gas fluxes, and plant and animal communities) for nearly 20 years to understand ecological and biological impacts from global changes and human disturbances. More details of the study region can be found in the Supplementary Information.

## Data sources

Long-term field data collected for a total of 53 different physical, chemical, and biological indicators of ecosystem services and over 11,000 empirical measurements were used in this study. The 53 datasets (i.e., 29 for grassland and 24 for wetland ecosystems) collected at BIR were grouped to quantify six categories of ecosystem services important to this region and grasslands in general[9], including soil nutrient maintenance, water quality regulation, carbon storage and greenhouse gas mitigation, biodiversity maintenance, invasion resistance of non-native species, and agricultural production. We strived to align a consistent set of indicators for the same ecosystem service assessed for both grasslands and wetlands. Temporal extent of data collection for individual indictors varied, but 50 out of the 53 datasets contained multi-year measurements, which were conducted between 2003 – 2020 (Tables 1, 2). Full details of ecosystem service indicators and data sources can be found in the Supplementary Information.

## Ecosystem service multifunctionality quantification

Prior to calculating ecosystem service multifunctionality indexes, we first standardized each indicator of ecosystem service to 0–1 range (Eq. 1) to remove effects of measurement scale differences among indicators, and also transformed certain indicators when necessary so that higher values always correspond to greater service provision. For example, because high water nutrients contribute to eutrophication (especially in our study system where nutrients are excessive), leading to algal blooms and habitat degradation, we thus transformed these indicators so that lower water nutrient concentrations represent greater water quality regulation service. Similarly, because lower greenhouse gas fluxes and invasive species diversity indicate greater provision of greenhouse gas mitigation and invasion resistance services, we transformed these indicators accordingly so that higher values mean greater service provision.

$$z_i = \frac{x_i - \min(x)}{\max(x) - \min(x)} \tag{1}$$

Based on scaled values of individual indicators (Tables 1, 2), we calculated ecosystem service multifunctionality using four different multifunctionality indexes (MFs), including simple averaging, quantile-based threshold, service-based weighted averaging, and cluster-based weighted averaging MFs[22,45,46]. We chose to include the simple averaging approach because it provides a comprehensive quantification of all available indicators and has been extensively used in many studies[45,82,83]. However, a simple averaging MF index might overweight categories of ecosystem services with a greater number of indicators or highly correlated indicators. Therefore, we included the service- and cluster-based weighted averaging approaches to avoid these potential biases[22,84]. Nevertheless, all averaging approaches could be affected by outliers, thus we further included the quantile-based threshold MF, which counts the number of indicators exceeding a quantile threshold and reduces the influence of extreme values[46,85]. As indicated above, different MF indexes have their advantages and disadvantages, and

encompassing diverse MF indexes can strengthen the robustness of our results.

In essence, the simple averaging MF was calculated by taking the unweighted average of all available indicators[45]. The threshold MF was determined as the number of indicators that achieved a 50th quantile threshold[46]. The service-based weighted averaging MF was calculated to avoid overweight indicators within the same ecosystem service category. Each service category was assigned weight coefficient as 1, then each indicator's weight was calculated using 1 divided by the numbers of indicators in that category (Eq. 2).

$$MF_{serivce-weighted} = \frac{1}{s} \times \sum_{i=1}^{n} ws_i \times s_i \tag{2}$$

where $s$ was the number of ecosystem services being categorized; $n$ represented the number of all indicators; $ws_i$ represented the weight coefficient calculated as 1 divided by the number of indicators in the corresponding service category; and $s_i$ was the scaled values of indicator $i$.

Similar to the service-based averaging approach, the cluster-based averaging MF index was calculated to down-weight statistically highly correlated indicators. To calculate this MF index, we firstly performed a hierarchical clustering analysis on all indicators and obtained six and seven clusters for grassland and wetland datasets, respectively (Fig. S3). Then, each cluster was assigned a weight coefficient of 1, and each indicator's weight was calculated using 1 divided by the numbers of indicators within each cluster (Eq. 3).

$$MF_{cluster-weighted} = \frac{1}{c} \times \sum_{i=1}^{n} wc_i \times s_i \tag{3}$$

where $c$ was the number of clusters classified by hierarchical clustering analysis; $n$ represented the number of all indicators; $wc_i$ represented the weight coefficient calculated as 1 divided by the number of indicators in its classified cluster; and $s_i$ was the scaled values of indicator $i$.

## Statistical analyses

To address our first and second questions, we tested effects of land-use intensification on each ecosystem service indicator using linear mixed-effects models, in which land-use intensity was treated as a fixed factor, and sampling year or season was treated as the random factor. To eliminate the influence of confounding factors such as geographic gradient in physiochemical properties, we also included elevation of sampling locations as a covariate. Transformations (i.e., log, square root, or cubic root) of response variables were performed when residuals of raw data failed to satisfy assumptions of linear regressions. Detailed model specification and goodness of fit for each individual ecosystem service indicator can be found in Table S3. To compare effects of land-use intensification across indicators, we calculated Hedge's $D$ as the standardized effect size and its 95% confidence intervals following Werling et al., (2014). We pooled standardized effect size of indicators within each category of ecosystem services and used average values to indicate the overall land-use intensification effect on that category of ecosystem service (Fig. 4).

To address our third question, we performed Kruskal-Wallis tests to analyze effects of land-use intensification on MFs of spatially coupled grasslands and wetlands. To address our last question on whether there were management-driven or intrinsic relationships (i.e., tradeoffs or synergies) among biophysical indicators of ecosystem services, we examined all available pairs of indicators across service categories using scattered plots and quantified differences in their relationships using Spearman's rank correlations, which are relatively robust to outliers and linear assumptions. Datasets of different indicators contained a mixture of single-time and repeated measurements. For those paired indicators with repeated measurements within same time

periods, indicator values were averaged and matched by measuring time and locations (either in grassland or wetland). For other indicators with single-time measurement, values were only averaged and matched by measuring locations. All statistical analyses were performed in R version 4.0.4[86]. Linear mixed-effects models were fitted using the "lme4" package[87].

## Reporting summary
Further information on research design is available in the Nature Portfolio Reporting Summary linked to this article.

## Data availability
All data used in this study is made publicly available and can be downloaded in the open repository Figshare with https://doi.org/10.6084/m9.figshare.24572368. Source data are provided with this paper.

## Code availability
All analyses were performed in R statistics, version 4.0.4, and all code to reproduce analyses is made publicly available and can be downloaded in the open repository Figshare with https://doi.org/10.6084/m9.figshare.24572368.

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

## Acknowledgements

We thank volunteers, interns and graduate students who helped in the collection and organization of these data. This research was funded by U.S. Department of Agriculture (USDA) National Institute of Food and Agriculture (NIFA) NRS project (FLA-FTL-005640; FLA-FTL-006277), McIntire-Stennis (FLA-FTL-005673), AFRI Foundational and Applied Science Program (2021-67013-33617), U.S. Department of Energy (DOE) Center for Advanced Bioenergy and Bioproducts Innovation (Office of Science, Office of Biological and Environmental Research under Award Number DE-SC0018420), Arizona State University (AZ, USA; Award No. ASU092762), Texas A&M AgriLife (Hatch project #9449), USDA NIFA (Project No. 2016-67019-24988), and a USDA cooperative agreement with Archbold (58-0202-7-001). Wetland data were collected with funding support from the U.S. EPA (STAR #RD-83456701-0) and USDA (CSREES #2006-35101-17204). Grassland vertebrate biodiversity was collected with Cooperative Agreements from the USDA – VS and USDA – NWRC (14-9200-0391 through 19-9200-0391). This research was a contribution from the Long-Term Agroecosystem Research (LTAR) network that is supported by USDA.

## Author contributions

JQ, EHB, YG, and HS conceived the ideas for this manuscript and defined the final analyses. EHB coordinated with data collection and preparation, and YG conducted the data analyses. YG and JQ designed and developed all the figures. YG and JQ wrote the first version of the paper. All authors contributed to the original data used in this manuscript, and commented on the manuscript.

## Competing interests

The authors declare no competing interests.

## Additional information

**Peer review information** : *Nature Communications* thanks the anonymous reviewers for their contribution to the peer review of this work. A peer review file is available.

[1]School of Forest, Fisheries, and Geomatics Sciences, Fort Lauderdale Research and Education Center, University of Florida, 3205 College Ave, Davie, FL, USA. [2]Archbold Biological Station, Buck Island Ranch, 300 Buck Island Ranch Road, Lake Placid, FL, USA. [3]School of Forest, Fisheries, and Geomatics Sciences, University of Florida, Gainesville, FL, USA. [4]U.S. Department of Agriculture, ARS Global Change and Photosynthesis Research Unit, Urbana, IL, USA. [5]Department of Biology, University of Central Florida, Orlando, FL, USA. [6]Department of Plant Biology, University of Illinois at Urbana – Champaign, Urbana, IL, USA. [7]Texas A&M AgriLife Research Center, Texas A&M University, Vernon, TX, USA. [8]Rangeland, Wildlife & Fisheries Management Department, Texas A&M University, College Station, TX, USA. [9]U.S. Department of Agriculture, APHIS Veterinary Services, Center for Epidemiology and Animal Health, Fort Collins, CO, USA. [10]Department of Ecology and Evolutionary Biology, Cornell University, Ithaca, NY, USA. [11]School of Natural Resources and Environment, University of Florida, Gainesville, FL, USA. ✉e-mail: eboughton@archbold-station.org; qiuj@ufl.edu

