## [Peer Review File · Nature Communications]

Reviewers' Comments:

Reviewer #1:

Remarks to the Author:

Comments to the Corresponding author

The manuscript "Agricultural intensification alters multifunctionality of metaecosystems" examines the long term impacts of two land management intensities, semi-natural and intensive managed grasslands, on ecosystem services in a subtropical, grassland-dominated landscape in Florida, USA. The results show that intensification increases yield which trades-off with the supply of multiple ecosystem services, highlighting the importance of balancing agricultural productivity with environmental conservation through sustainable intensification or similar practices that promote multiple ecosystem services. Further, the study reveals a cascading effect from grassland intensification to wetlands, indicating that land intensification can impact the supply of ecosystem services in natural wetlands embedded within grassland, which can weaken the multifunctionality of grassland-wetland metaecosystems.

The study is innovative by including a meta-ecosystem framework to assess the impacts of land intensification on ecosystem services and its focus on examining trade-offs and synergies in intensification that enable land use mosaics. The study also takes a holistic and landscape perspective, aiming to balance agricultural productivity with environmental conservation. Finally, the study's findings highlight the complementary nature of different land management intensities (i.e., SN and IM) and suggest that these two management intensities should be implemented together and spatially optimized at the landscape scale to achieve sustainable intensification.

The study provides a unique showcase example of how holistic perspectives can help disentangle interacting and cascading effects and improve our understanding of how land use intensification affects the multifunctionality of ecosystems. The manuscript appears to be very well written, structured and presented in a way that makes it accessible to readers with a range of backgrounds and expertise. The chosen meta-analytical analysis approach is appropriate and well implemented.

Main only concern with the manuscript is that it does not (or only very briefly) address some of the study's limitations in the discussion. For example:

the study focuses on a specific region and may not be generalizable to other areas - even with similar environmental conditions and farming practices (as suggested in the introduction Australia and South America). It would be interesting to learn about potential hindrances for generalisations to other subtropical regions.

the study does not investigate agricultural yield (e.g. fodder/meat harvested) and by extension the economic implications of the two intensities and how implementing landscape-level meta-ecosystem strategies for intensification would affect trade-offs between income and other ecosystem services. I am aware that detailed yield/income data is hard to come by and I am not criticising the authors for not including that here. But since financial outcomes are one of the main drivers for decision making in the agricultural context it would be interesting to include that perspective in the discussion.

the study does not consider the potential impact of climate change on ecosystem functions and how it may interact with agricultural intensification. Again, not a critique for not addressing this directly but could be mentioned as a limitation/confounding effect.

The lack of these aspects in the discussion does not reduce the value of this study, but including (some of) these aspects could help provide a more nuanced and reflected view.

However, overall these are very minor concerns and I think that the study addresses an important and timely topic and will surely be a valuable contribution to the existing body of literature. Therefore, I recommend it for publication.

Minor:

Line 325: bookmark error disrupts text

Reviewer #2:

Remarks to the Author:

Manuscript NCOMMS-23-11027

Comments to the Authors (Guo et al. "Agricultural intensification alters multifunctionality of metaecosystems")

This manuscript could be an interesting contribution to ecosystem service literature that uses long-term data set of multiple ecosystem variables (some pools, some functions, some services) measured from a heterogeneous mosaic of three different types of ecosystems: seminatural grasslands, intensively used grasslands, and wetlands that are embedded in the two grassland types. The authors find that more intense grassland management results in high nutrient concentrations, higher quality plant biomass and livestock production, but greater exotic species richness, and lower plant and vertebrate diversities in grasslands. These effects were reflected in wetlands that show, for example, lower water quality and higher CH₄ emissions, when embedded in intensively used grasslands than when embedded in seminatural grasslands.

General comments

These results are definitely interesting but to some extent confirm previous knowledge. For example, it is well known that intense land-use (including fertilization) can result in increased soil nutrients and productivity, and thus increased forage quality and quantity, while at the same time decrease biodiversity and increase exotic invasions (e.g., Allan et al. 2015 *Ecol Lett*, Borer et al. 2014 *Nature*, Eskelinen et al. 2020 *Ecology*). Many papers show that grazing could counteract these effects, but the authors do not discuss the possibly opposing effects of their management practices. Further, some results seem very trivial: if intensive management involved removal of natives and addition of exotics (as I understood from the Discussion), one would of course expect decrease in native diversity. It is hard to understand these results as information concerning which management practices were included is given nowhere in the text. This is essential information without which it is impossible to understand the results, and it has to be in the manuscript itself, not just in the supplementary.

The most interesting and novel part of the manuscript is how the land-use effects on grasslands cascade to wetlands that are embedded in grasslands. These systems mostly seem to represent resource-flow based meta-ecosystems (Gounand et al. 2018 *TREE*, Figure 1) where adjacent, different kinds of ecosystems are connected via resource flows crossing their boundaries, and true dispersal is missing. In this manuscript, flow of resources, though, seems to happen only to one direction, i.e., from grasslands to wetlands. Or at least that is how the authors approach their system but which could also be prove incorrect. Overall, the whole meta-ecosystem perspective seems superficially presented, is poorly founded, and fits the data poorly. I suggest that the authors either omit referring to meta-ecosystem framework completely and focus on the result that land-use and management in one ecosystem can reflect to another, or focus on resource-flow based meta-ecosystem theory and consider resource-flow reciprocally between the ecosystems (i.e., how wetlands affect surrounding grasslands and the other way round).

Overall, the manuscript is overly complexly written, and trendy jargon (e.g., meta-ecosystem framework) is often used without saying much concrete. The language and message of the manuscript should be simplified, focused, and clarified. Some parts of the manuscript are also overly long; for example, Discussion is almost 7.5 pages long, and could be substantially shortened (by several pages). Sometimes it is not clear what the authors are studying as ecosystem services and functions are used in a confusing way. There are also many other issues, e.g., missing information in Results and Methods, and inconsistencies between results and discussion, that are addressed in my detailed comments. It would be great if the authors clarified these issues.

I really appreciate the authors' attempt to put lots of valuable data together to address how land-

use intensification affects ecosystem functions/services of grasslands and wetlands embedded in them, which is an important topic, but data alone does not justify publication and presentation should be simplified/clarified to make the manuscript more appealing.

Detailed comments

Abstract

- Line 44: What is "...ecosystem service interaction"? This is a good example of complex language that is really unclear.
- Line 46: What does "metaecosystems with similar magnitude" mean? The same problem than above.
- Lines 36-38 and 46-48: How did you define and measure "sustainability" of land-use in your system? This sounds something that was not part of your study at all and is pure speculation.

Introduction

- Line 57: "an important" instead of "a main".
- Line 79: Why "hidden"?
- Lines 89-91: See Allan et al. 2015 Ecol Lett, Neyret et al. 2023 Nat Sus, Le Provost et al. 2023 Nat Ecol Evol
- Line 106: "...integrate and interact...".
- Paragraph lines 105-115: Here you could develop this further to talk about resource-flow based meta-ecosystems based on Gounand et al. 2018 TREE, Figure 1.
- Line 125: "In Florida, this has..." -> what "this"?
- In general, please explain what does intense land-use mean here? How did SN and IM grasslands differ? I think it is important to describe these to some extent in the Introduction, otherwise it is impossible to understand the results and discussion.
- Introduction is pretty long, could be shortened.
- Since this is said to be a meta-ecosystem, one expects to see many grasslands and wetlands with different connectivity to each other and flow of materials/energy. The description should be clearer here in order for the reader to understand the system and results/discussion.
- "Land-use intensification" rather than "land intensification" throughout the manuscript.

Results

- Lines 156-157: What does intensified management mean in this system? Has to be explained, otherwise not possible to understand the results.
- Line 160: These are nutrient and carbon pools, not functions.
- Line 162: β -diversity among grasslands or wetlands or between them? Diversity of vascular plants?
- Lines 161-166: Is this all in grasslands? In general, it is often difficult to know whether the authors talk about grasslands or wetlands.
- Lines 171-172: Not clear, please rephrase.
- Line 174: Why does greater total N and P equal lower quality? I understand what you mean but it should not be considered self-evident.
- Lines 170-180: "Grassland intensification also reduced" – this sounds causal, like if you looked at temporal change in response to altered management practices. Is this the case? Or should this rather be "Wetlands embedded in IM grasslands had lower plant and ectothermic vertebrate diversities compared to wetlands embedded in SN grasslands"? Please check throughout the results and discussion for similar expressions and clarify.
- Line 185: Please mention the approaches here or at least provide references. MF approaches are quite different and can produce different results, so it matters which ones you used.
- Lines 192-195: Something wrong here, please rephrase.
- Line 198: Delete "synergies" (I guess it's just correlations here?).

Discussion

- In general, Discussion is too long, can be shortened by several pages.
- Line 217: This is not a synthesis.
- Lines 217-222: Overly long and complicated sentence, please rephrase/simplify.
- Line 238: What does reduced water quality mean here? High P concentration? Why does that mean reduced value?

- Lines 240-243: This is against theory that predicts that plants should invest relatively more on aboveground biomass when nutrients are not limiting. Furthermore, your Results section says "Compared to SN wetlands, IM wetlands had lower root biomass, but higher above-ground primary productivity" (lines 176-177) – there is apparent inconsistency here.
- Line 145 and 248-250: Some of your specific management types are such that they could have an opposite impact on biomass and richness. Fertilization increases productivity and competition for light, while grazing removes biomass and can increase plant diversity by alleviating competition for light, and could counteract the negative effects of fertilization on diversity (Hautier et al. 2009, Eskelinen et al. 2022).
- Line 247: What treatments are these? If native plants were removed actively by humans and this removal is coupled with seeding of exotics, it is no wonder that diversity, specifically that of natives, decreases. If intensive management involves these, then the results are trivial.
- Line 251: But if you removed native species, then that's the likeliest reason their diversity to decrease.
- Line 253: Implies causality, please rephrase.
- Line 266: Were wetlands also grazed?
- Lines 266-270: How can you separate the effects of grazing from the other management practices? If several factors are manipulated at the same time (e.g., grazing, fertilization, native reduction, exotic addition etc.), there is no way to tease apart their effects. Also, if this is based on correlation, then you can't imply causality ("increased"): please use wording like "was associated with" or similar. Please check and rephrase throughout discussion.
- Lines 282-310: This is otherwise great but is a way beyond what was studied in this study.
- Lines 314-316: This is an important result and the authors could highlight this more in the light of Gounand et al. 2018 TREE.
- Line 325: Something extra here.
- Line 332: Was assessing trade-offs based on correlations?
- Lines 352-353: This is nicely written – I would highlight this result more throughout the manuscript.

Methods

- Line 390: Numerous -> how many exactly?
- In general, how did you define whether lower or higher value was desirable for each service/function?
- Where is the information about which management practices were involved and applied to the grasslands? Should be in the manuscript text.

Reviewer #1 (Remarks to the Author):

Comments to the Corresponding author

The manuscript “Agricultural intensification alters multifunctionality of metaecosystems” examines the long term impacts of two land management intensities, semi-natural and intensive managed grasslands, on ecosystem services in a subtropical, grassland-dominated landscape in Florida, USA. The results show that intensification increases yield which trades-off with the supply of multiple ecosystem services, highlighting the importance of balancing agricultural productivity with environmental conservation through sustainable intensification or similar practices that promote multiple ecosystem services. Further, the study reveals a cascading effect from grassland intensification to wetlands, indicating that land intensification can impact the supply of ecosystem services in natural wetlands embedded within grassland, which can weaken the multifunctionality of grassland-wetland metaecosystems.

The study is innovative by including a meta-ecosystem framework to assess the impacts of land intensification on ecosystem services and its focus on examining trade-offs and synergies in intensification that enable land use mosaics. The study also takes a holistic and landscape perspective, aiming to balance agricultural productivity with environmental conservation. Finally, the study's findings highlight the complementary nature of different land management intensities (i.e., SN and IM) and suggest that these two management intensities should be implemented together and spatially optimized at the landscape scale to achieve sustainable intensification.

The study provides a unique showcase example of how holistic perspectives can help disentangle interacting and cascading effects and improve our understanding of how land use intensification affects the multifunctionality of ecosystems. The manuscript appears to be very well written, structured and presented in a way that makes it accessible to readers with a range of backgrounds and expertise. The chosen meta-analytical analysis approach is appropriate and well implemented.

Thank you very much for the nice summary of this work, as well as your positive feedback and comments on our manuscript.

Main only concern with the manuscript is that it does not (or only very briefly) address some of the study's limitations in the discussion. For example:

the study focuses on a specific region and may not be generalizable to other areas - even with similar environmental conditions and farming practices (as suggested in the introduction Australia and South America). It would be interesting to learn about potential hindrances for generalisations to other subtropical regions.

the study does not investigate agricultural yield (e.g. fodder/meat harvested) and by extension the economic implications of the two intensities and how implementing landscape-level meta-ecosystem strategies for intensification would affect trade-offs between income and other ecosystem services. I am aware that detailed yield/income data is hard to come by and I am not

criticising the authors for not including that here. But since financial outcomes are one of the main drivers for decision making in the agricultural context it would be interesting to include that perspective in the discussion.

the study does not consider the potential impact of climate change on ecosystem functions and how it may interact with agricultural intensification. Again, not a critique for not addressing this directly but could be mentioned as a limitation/confounding effect.

We fully agree with the reviewer that these are great points to consider and discuss and include as our study limitations, which will further help allude to future research needs.

In our revision, we have now largely modified our Discussion section, and added a new paragraph towards the end of Discussion to discuss these potential limitations, including (1) the extent to which our findings can be generalizable to other tropical and subtropical biomes and other agricultural-wetland metaecosystems where scales, environmental contexts, and farming practices can vary. This is in particular relevant for grassland ecosystems that are often managed under a portfolio of different and integrated practices. Hence, different biotic and abiotic factors along with combinations of practices could lead to different outcomes due to land-use intensification. (2) the need of considering other drivers of global changes and anthropogenic pressures such as climate change and how it interacts and potentially confounds with agricultural intensification to alter the provision of ecosystem services and multifunctionality of metaecosystems. Please see Lines 346-362 in our revised manuscript.

In addition, the reviewer also pointed out the importance of considering agricultural yields (e.g., fodder/meat harvested) and associated economic implications from the two land-use intensities, which we agree wholeheartedly. We also appreciate the reviewer's recognition of challenges to consistently obtain such yield and economic datasets, which is why we haven't included them in our formal analyses. But it is crucial to address the degree to which the implementation of landscape-level metaecosystems strategies for intensification would affect tradeoffs between income/financial returns and other ecosystem services. In our studied system, historical livestock calf-cattle production data has been collected at the aggregated ranch scale, along with the economic model work that has been performed and published. Hence, to address this comment, we have now included relevant financial outcomes and economic impacts associated with management practices in our Discussion to better illustrate and characterize differences between SN and IM practices. Inclusion of such information in our revision could help broaden the relevance of our work and can be critical for decision-making and landscape management in the agricultural context. Please see Lines 285-298 in the revised manuscript.

The lack of these aspects in the discussion does not reduce the value of this study, but including (some of) these aspects could help provide a more nuanced and reflected view.

However, overall these are very minor concerns and I think that the study addresses an important and timely topic and will surely be a valuable contribution to the existing body of literature. Therefore, I recommend it for publication.

Thanks again for your positive feedback and suggestions.

Minor:

Line 325: bookmark error disrupts text

Thanks for pointing this out due to glitches in the citation software. Bookmark error is now being removed in the revision.

Reviewer #2 (Remarks to the Author):

Manuscript NCOMMS-23-11027

Comments to the Authors (Guo et al. “Agricultural intensification alters multifunctionality of metaecosystems”)

This manuscript could be an interesting contribution to ecosystem service literature that uses long-term data set of multiple ecosystem variables (some pools, some functions, some services) measured from a heterogeneous mosaic of three different types of ecosystems: seminatural grasslands, intensively used grasslands, and wetlands that are embedded in the two grassland types. The authors find that more intense grassland management results in high nutrient concentrations, higher quality plant biomass and livestock production, but greater exotic species richness, and lower plant and vertebrate diversities in grasslands. These effects were reflected in wetlands that show, for example, lower water quality and higher CH₄ emissions, when embedded in intensively used grasslands than when embedded in seminatural grasslands.

Thank you for your summary of our work.

General comments

These results are definitely interesting but to some extent confirm previous knowledge. For example, it is well known that intense land-use (including fertilization) can result in increased soil nutrients and productivity, and thus increased forage quality and quantity, while at the same time decrease biodiversity and increase exotic invasions (e.g., Allan et al. 2015 Ecol Lett, Borer et al. 2014 Nature, Eskelinen et al. 2020 Ecology). Many papers show that grazing could counteract these effects, but the authors do not discuss the possibly opposing effects of their management practices. Further, some results seem very trivial: if intensive management involved removal of natives and addition of exotics (as I understood from the Discussion), one would of course expect decrease in native diversity. It is hard to understand these results as information concerning which management practices were included is given nowhere in the text. This is essential information without which it is impossible to understand the results, and it has to be in the manuscript itself, not just in the supplementary.

We much appreciate the positive comments and feedback on our manuscript, and for pointing out additional studies that are relevant to this work, which we have now included these

references in our revision. The reviewer raised a very good point that intensive management involved removal of natives and addition of exotics (e.g., more productive forage grass species), which would lead to seemingly ‘intuitive’ results with lower native diversity and higher non-native diversity under high-intensity management. However, in our dataset, we also found out that several of these non-native species in intensively managed grasslands were not from intentional planting, such as Sporobolus indicus, Panicum repens, Cyperus brevifolia, Murdannia nudiflora, to name just a few. Hence, the revealed differences in non-native plant diversity between IM and SN grasslands could be due to not only planting and introduction of exotic species, but also species interactions in the grassland communities. One possible explanation could be related to the ‘invasional meltdown’ (Simberloff and Von Holle 1999), where successful exotic/invasive species may facilitate the establishment and success of others, thus resulting in communities less resistant to invasion. In addition, for embedded wetlands, our results (Fig. 1E vs. Fig. 2F) showed even stronger effects of land-use intensification on increasing exotic plant species richness (i.e., indicative of native suppression and invasion dispersal), where natives removal and forage grass planting were not practiced. To address this comment, we have revised our discussion to clarify these points (please see Lines 248 – 259 in the revised manuscript).

In addition, we also thank the reviewer for the great suggestions of including the management practices in the Main text to help better understand and interpret the results. Thus, in our revised manuscript, we have now added a brief description of management practices in the Introduction (see Lines 126-130), and also more detailed explanations in the Methods section (please see Lines 412-425 in the revised manuscript).

The most interesting and novel part of the manuscript is how the land-use effects on grasslands cascade to wetlands that are embedded in grasslands. These systems mostly seem to represent resource-flow based meta-ecosystems (Gounand et al. 2018 TREE, Figure 1) where adjacent, different kinds of ecosystems are connected via resource flows crossing their boundaries, and true dispersal is missing. In this manuscript, flow of resources, though, seems to happen only to one direction, i.e., from grasslands to wetlands. Or at least that is how the authors approach their system but which could also be prove incorrect. Overall, the whole meta-ecosystem perspective seems superficially presented, is poorly founded, and fits the data poorly. I suggest that the authors either omit referring to meta-ecosystem framework completely and focus on the result that land-use and management in one ecosystem can reflect to another, or focus on resource-flow based meta-ecosystem theory and consider resource-flow reciprocally between the ecosystems (i.e., how wetlands affect surrounding grasslands and the other way round).

We much appreciate the reviewer’s positive feedback on the novelty of this research and recognition of its relevance to the metaecosystems framework (Gounand et al. 2018 TREE), where adjacent different ecosystems (i.e., coupled grasslands and wetlands here in this context) are spatially connected via resource flows, organismal movements, and other ecological or biophysical processes across system boundaries.

We acknowledge and fully agree with the reviewer that the fit of our empirical datasets to the metaecosystems framework could be further improved, especially if better quality of data (e.g., specific quantification of spatial flows) were available. However, as also pointed out in Gounand

et al. (2018), despite the theoretical development of the metaecosystems framework, adoption by empiricists has generally lagged. This is primarily due to the high level of abstraction of the theory and challenges in collecting and assembling consistent, dynamic, and actual spatial flow datasets (e.g., studies thus far are still dominated by dispersal and nutrient transport). This is particularly true for our study, in which we strived to integrate holistic and systematic perspectives with the metaecosystems framework to understand land-use intensification effects on the multifunctionality of grasslands-wetlands with 53 different indicators. Consistently measuring the different spatial and reciprocal flows between grasslands and wetlands underlying all these indicators over the long-term (i.e., ~ two decades here) could be extremely valuable to improve understanding of their spatial dynamics, but is a daunting task to achieve.

Nevertheless, with all due respect and in consideration of the comments from Reviewer #1, we still recognized the value of the metaecosystems framework in this research and more broadly in the context of holistic agricultural and landscape management, where local intensification (or other agricultural practices) can exert spatial cascading effects on adjoining natural ecosystems, thus compromising multifunctionality of metaecosystems beyond production areas. Such effects were often neglected, not well quantified, and rarely accounted for in the agricultural management and decision-making. Hence, this work can serve as a novel and firsthand study to illustrate the usefulness of the metaecosystems framework and importance of spatially displaced impacts, and as a call for more in-depth future measurements and investigations of spatial flows within the metaecosystems for sustainable agricultural intensification.

With that said, in our revision, we have made substantive changes to address this comment. Specifically, we have: (1) toned down and reduced the metaecosystems language wherever possible in describing our results throughout the manuscript; (2) clarified that our uses and references to the metaecosystems framework are primarily centered on the spatially connected ecosystems and consequences of spatial cascading effects for ecosystem functions and services and multifunctionality; (3) acknowledged in the newly added limitation section of Discussion that our results were the consequences of land-use intensification on multiple ecosystem services, as manifested through the spatial flows within the metaecosystems, which were inferred but not explicitly or directly measured in our study; and (4) further added in the study limitation that the metaecosystems framework (regardless of dispersal- or resource-flow based metaecosystems) should also consider reciprocal spatial flows between and among ecosystems.

While it may seem ‘intuitive’ that land-use intensification effects on many of our measured responses (e.g., hydrological, nutrient flows) are likely to be dominated by the directional flows from grasslands to wetlands, due to physical geography and how these processes might work laterally across the landscapes, we agree with the reviewer that other biotic processes and resource flows from wetlands to grasslands are plausible and likely occur. Examples include forage for cattle and subsidies in the form of aquatic life that support upland food webs, or cattle foraging in wetlands that further transport nutrients to upland grasslands, and help disperse wetland species in other distant and isolated wetlands and ditches. These are fruitful avenues for future research to provide further empirical evidence supporting the metaecosystems theory development, especially pertinent to understanding, e.g., how do different and reciprocal spatial flows within the metaecosystems interact, and at which spatial scales such interactions are most dominant to determine the consequences for ecosystem services and multifunctionality of

metaecosystems. Please see examples of our major revisions in Lines 108-113, 226-233, 312-314, 316-318, and 362-376.

Overall, the manuscript is overly complexly written, and trendy jargon (e.g., meta-ecosystem framework) is often used without saying much concrete. The language and message of the manuscript should be simplified, focused, and clarified. Some parts of the manuscript are also overly long; for example, Discussion is almost 7.5 pages long, and could be substantially shortened (by several pages). Sometimes it is not clear what the authors are studying as ecosystem services and functions are used in a confusing way. There are also many other issues, e.g., missing information in Results and Methods, and inconsistencies between results and discussion, that are addressed in my detailed comments. It would be great if the authors clarified these issues.

Thank you for pointing out these shortcomings. In our revision, we have now simplified our language, reduced the use of technical jargons as much as possible, and clarified the messages based on reviewer's detailed comments below. We have condensed the Discussion to 6.5 pages (from the original draft of 7.5 pages) in the revision, even with the addition of a separate new paragraph on study limitations, per suggestions from the reviewers.

In addition, we have also made edits throughout the manuscript to clarify our consistent uses of terminologies such as ecosystem functions and services (e.g., please see Lines 458-468) and the metaecosystems (e.g., see Lines 42-43, 108-113). We have also added additional information as suggested in the Methods (e.g., Lines 412-425), and thoroughly revised the Results section to improve consistent and accurate presentations (please examples of our revisions in Lines 160-181, 186-190, 211-214, and 220-223 in the revised manuscript). Any missing information has also been added to the corresponding sections in the revised manuscript. Please see all detailed changes explained below.

I really appreciate the authors' attempt to put lots of valuable data together to address how land-use intensification affects ecosystem functions/services of grasslands and wetlands embedded in them, which is an important topic, but data alone does not justify publication and presentation should be simplified/clarified to make the manuscript more appealing.

We are extremely grateful for all your constructive and insightful comments on our manuscript, which has been really instrumental for us to revise and strengthen this work with a clearer picture of such complex, multi-dimensional, and long-term datasets of coupled grassland-wetland ecosystems. We have carefully considered and incorporated all your thorough and detailed suggestions and feedback in our revision, which we believe has largely improved the clarity and robustness of our work. As explained above and detailed below, we have now made substantial revisions to address these concerns and improved the presentation of this manuscript so that our findings can be more appealing and relevant to sustainable landscape management and agricultural intensification.

Detailed comments

Abstract

- Line 44: What is "...ecosystem service interaction"? This is a good example of complex language that is really unclear.

We have revised the wording to make it clearer. Please see Line 49 in the revised manuscript.

- Line 46: What does "metaecosystems with similar magnitude" mean? The same problem than above.

Sorry for the confusion. Here we were trying to characterize that multifunctionalities of managed grasslands and embedded wetlands were affected to the similar extent by upland land-use intensification. We have modified our phrase to clarify in the revision (please see Line 51).

- Lines 36-38 and 46-48: How did you define and measure "sustainability" of land-use in your system? This sounds something that was not part of your study at all and is pure speculation.

This is a good point. In the Abstract, we meant to introduce the concept of "sustainable intensification" (Pretty et al. 2018), which is a specific term referring to the transition of agricultural systems to simultaneously achieve productivity, ecosystem services and multifunctionality. The reviewer is correct that we did not measure 'sustainability' per se in this work, and our focus here is multifunctionality. To avoid confusion, we have modified our Abstract accordingly (please see Line 53 in the revision).

Introduction

- Line 57: "an important" instead of "a main".

Done. See Line 62.

- Line 79: Why "hidden"?

The word 'hidden' is now revised. See Lines 82-83 of the revised manuscript.

- Lines 89-91: See Allan et al. 2015 Ecol Lett, Neyret et al. 2023 Nat Sus, Le Provost et al. 2023 Nat Ecol Evol

Thank you very much for suggesting these great relevant literature. We have revised this paragraph and updated our citations. Please see Lines 92-98 in the revision.

- Line 106: "...integrate and interact...".

Done. See Line 106 in the revised manuscript.

- Paragraph lines 105-115: Here you could develop this further to talk about resource-flow based meta-ecosystems based on Gounand et al. 2018 TREE, Figure 1.

Thank you for your great suggestions. We have developed this further to include discussions of spatial flows in the grassland-wetland metaecosystems, based on Gounand et al. (2018). We have included examples of different spatial and resource flows, which can occur in the spatially connected grassland-wetland mosaic. Please see Lines 108-113 in our revised manuscript.

- Line 125: “In Florida, this has...” -> what “this”?

Revised. See Lines 133-134.

- In general, please explain what does intense land-use mean here? How did SN and IM grasslands differ? I think it is important to describe these to some extent in the Introduction, otherwise it is impossible to understand the results and discussion.

Great suggestion. Please see our responses above, and in brief, in the revision, we have now added additional explanations on the typical practices of land-use intensification in the tropical and subtropical grasslands, as well as the specific distinctions between SN and IM grasslands in their implemented integrated practices in our study context. Please see Lines 126-130 and Lines 412-425 in the revised manuscript.

- Introduction is pretty long, could be shortened.

Thanks for pointing this out. In our revision, we have simplified our language and condensed the Introduction wherever possible. During the revision, we have added additional content per reviewers’ suggestions. But even with these additions, we were also able to reduce the Introduction by ~0.5 page.

- Since this is said to be a meta-ecosystem, one expects to see many grasslands and wetlands with different connectivity to each other and flow of materials/energy. The description should be clearer here in order for the reader to understand the system and results/discussion.

Please see our response above on the descriptions and examples of spatial flows characterizing grassland-wetland metaecosystems. Also, many thanks for your earlier suggestions to discuss this in the context of resource-flow based metaecosystems.

- “Land-use intensification” rather than “land intensification” throughout the manuscript.

Thanks for the suggestions. We now have revised and used the term “land-use intensification” consistently throughout the manuscript. Please see highlighted changes in the revision.

Results

- Lines 156-157: What does intensified management mean in this system? Has to be explained, otherwise not possible to understand the results.

As included in the responses above, to help better understand and interpret the results, we have now added descriptions on integrated management practices under intensified land uses in both Introduction (see Lines 126-130), and more details in the Method section (see Lines 412-425).

- Line 160: These are nutrient and carbon pools, not functions.

Good point. We agree with the reviewer that some of our included measurements are ecosystem pools, and not functions. However, changes in these pools are the consequences of land-use intensification, which can result from or further affect functions such as nutrient retention and carbon cycling and thereby service provision. These are commonly used indicators for ecosystem services and for quantifying multifunctionality (e.g., Felipe-Lucia et al. 2018; Qiu et al. 2019; Tamburini et al. 2022). However, one improvement in our study is that, to ensure the robustness of our results, instead of using individual indicators, we used multiple relevant indicators to characterize the provision of one ecosystem service (see Table S2 and S3), which was then used to measure multifunctionality. To avoid confusion, wherever possible, we clarified that not all of our measurements are functions in the revised manuscript.

- Line 162: □-diversity among grasslands or wetlands or between them? Diversity of vascular plants?

Yes, here we referred to the vascular plants □-diversity among grasslands.

- Lines 161-166: Is this all in grasslands? In general, it is often difficult to know whether the authors talk about grasslands or wetlands.

Sorry for the confusion. Yes, the first paragraph of the Results is entirely about grasslands, and the second paragraph is solely about wetlands.

To further clarify, we have largely revised and reorganized this section to make it clearer. In addition, we also added subsection titles in the Result to further clarify how the results corresponded to each of our research question as outlined in the Introduction. Please see the updated Result section in the revision.

- Lines 171-172: Not clear, please rephrase.

We have now rephrased and revised these sentences to improve clarity. Please see Lines 178-181 in the revision.

- Line 174: Why does greater total N and P equal lower quality? I understand what you mean but it should not be considered self-evident.

We have added the justification in the Methods section to help clarify the use of biophysical indicators and their relevance to the provision of ecosystem services. Please see Lines 458-468 in the revised manuscript.

- Lines 170-180: “Grassland intensification also reduced” – this sounds causal, like if you looked

at temporal change in response to altered management practices. Is this the case? Or should this rather be “Wetlands embedded in IM grasslands had lower plant and ectothermic vertebrate diversities compared to wetlands embedded in SN grasslands”? Please check throughout the results and discussion for similar expressions and clarify.

This is an excellent point. In our study (and also as stated in our manuscript, e.g., Lines 153-156), we used the comparison between IM and SN grasslands and wetlands to infer the land-use intensification effects, instead of comparing the responses before and after intensification was applied. In our experimental design and data collection, we have strived to control for any potential confounding factors so that the differences between IM and SN grasslands and wetlands are primarily owing to land-use intensification effects. Nonetheless, this is an important distinction to make, and we have now revised our expressions and made thorough edits throughout the Results and Discussion sections to clarify and avoid any causal expressions.

- Line 185: Please mention the approaches here or at least provide references. MF approaches are quite different and can produce different results, so it matters which ones you used.

Good point. We have added the details and references for the MF approaches here. Please see Lines 194-197 in the revised manuscript.

- Lines 192-195: Something wrong here, please rephrase.

Revised. Please see Lines 203-205 in the revision.

- Line 198: Delete “synergies” (I guess it’s just correlations here?).

Done. See Line 209 and similar changes throughout the Result section for consistency.

Discussion

- In general, Discussion is too long, can be shortened by several pages.

Great suggestions. As responded above, we have now made substantive edits and revisions in the Discussion to shorten the text and also ensure that our finalized manuscript is within the length limit of the article type in the journal.

- Line 217: This is not a synthesis.

Revised throughout the manuscript.

- Lines 217-222: Overly long and complicated sentence, please rephrase/simplify.

We have simplified and rephrased the first paragraph of the Discussion. Please see Lines 226-233 in the revised manuscript.

- Line 238: What does reduced water quality mean here? High P concentration? Why does that mean reduced value?

We added the explanation and justification about the relationship between biophysical indicators and ecosystem services in the Method section (please see Lines 458-468 in the revision).

- Lines 240-243: This is against theory that predicts that plants should invest relatively more on aboveground biomass when nutrients are not limiting. Furthermore, your Results section says “Compared to SN wetlands, IM wetlands had lower root biomass, but higher above-ground primary productivity” (lines 176-177) – there is apparent inconsistency here.

Thanks for pointing this out. Apologies for the confusion. In our revision, we have revised here and removed these sentences based on other comments that it is challenging to disentangle the specific management practice implemented under the integrated IM practices and the need to simplify the Discussion (e.g., please see the second paragraph of the revised Discussion). We also double checked the results and made sure that all the presentations are consistent.

- Line 145 and 248-250: Some of your specific management types are such that they could have an opposite impact on biomass and richness. Fertilization increases productivity and competition for light, while grazing removes biomass and can increase plant diversity by alleviating competition for light, and could counteract the negative effects of fertilization on diversity (Hautier et al. 2009, Eskelinen et al. 2022).

Thank you for pointing out effects of fertilization and grazing from other studies, which are helpful to compare and contextualize. However, our pasture systems could be different in such dynamics, due to complete replacement of native plant community with bahiagrass that leads to different responses to fertilization and grazing. Specifically, our data and other work published in our study system at the BIR suggested that cattle grazing does not fully counteract the impact of other practices within intensified management (e.g., fertilization, pasture planting).

To address this comment, we have now added references of previous research findings from our study region about the specific grazing effects on plant diversity in the IM and SN grasslands and wetlands. Please see details in Lines 255-259 in the revised manuscript.

- Line 247: What treatments are these? If native plants were removed actively by humans and this removal is coupled with seeding of exotics, it is no wonder that diversity, specifically that of natives, decreases. If intensive management involves these, then the results are trivial.

We apologize for the confusion. It is true that the plant diversity loss in grasslands was primarily due to direct management of human removing natives and planting exotic forage grasses. However, as responded previously, not all identified exotic plant species are from intentional planting. Further, wetlands, in contrast to pastures, were not directly managed with native species removal or seeding of exotics. Thus, the plant diversity declines in wetlands embedded in improved pastures were possibly associated with persistent long-term nutrient flows, higher grazing pressure, and species dispersal from increased ditching and intensively managed

uplands (Boughton et al. 2010; Boughton et al. 2011). Please see our responses above to this comment. To address this comment, we have made revisions in Lines 248-255 to help clarify.

- Line 251: But if you removed native species, then that's the likeliest reason their diversity to decrease.

Thanks for pointing this out. Please see our response above on this and similar comment.

- Line 253: Implies causality, please rephrase.

Revised, and removed any possible causality inferences throughout the manuscript.

- Line 266: Were wetlands also grazed?

Yes, wetlands embedded in grasslands were exposed to the same grazing pressure and this was tracked at the pasture level. Cattle used wetlands for supplementary forages and heat regulation (Pandey et al. 2009).

- Lines 266-270: How can you separate the effects of grazing from the other management practices? If several factors are manipulated at the same time (e.g., grazing, fertilization, native reduction, exotic addition etc.), there is no way to tease apart their effects. Also, if this is based on correlation, then you can't imply causality ("increased"): please use wording like "was associated with" or similar. Please check and rephrase throughout discussion.

This is a great point, and is a typical phenomenon and challenge in grassland management and research, where a portfolio of specific practices (e.g., fertilization, drainage, grazing pressure, and conversion of native grasses into productive forages) were integrated and implemented simultaneously at the landscape scale under one type of land use. Hence, it is very difficult to disentangle and separate the effects of one practice without factorial experimental manipulations in the field. Even with experimental manipulations, it is highly complex if more than three practices are considered and analyzed, especially when there are possible interactions among these practices that could amplify or counteract each other. From a grassland management and multifunctionality standpoint, it is important to understand the effects of land-use intensification (as being implemented in the real-world ranch operations), where all these different practices are acting in concert.

Hence, to address this comment, in our revision, we have (1) removed and minimized any speculations regarding the single factor effects, as well as revised expressions that could imply causality; (2) wherever possible, we referenced previous and published studies in our study region of Buck Island Ranch that specifically tested and separated individual management practice to help explain our findings; and (3) included in the study limitation section a description of the importance of teasing apart the effects of multiple integrated management practices under land-use intensification (e.g., see Lines 354-357 in the revision).

- Lines 282-310: This is otherwise great but is a way beyond what was studied in this study.

We have largely modified and condensed this section in the Discussion, and instead focused on statements with data support (e.g., addition of economic data per comments from Reviewer #1). Please see Lines 285-310 in the revised manuscript.

-Lines 314-316: This is an important result and the authors could highlight this more in the light of Gounand et al. 2018 TREE.

Thank you for your suggestions. In our edits throughout the manuscript, we have strived to highlight more on this point in the light of Gounand et al. (2018).

- Line 325: Something extra here.

Removed.

- Line 332: Was assessing trade-offs based on correlations?

Yes, we assessed trade-offs based on correlations.

- Lines 352-353: This is nicely written – I would highlight this result more throughout the manuscript.

Thanks for your positive feedback and suggestions. In our revision and edits throughout the manuscript, we have highlighted this result more, and also emphasized that our use of the metaecosystems framework is as stated here focused on the direct and spatial cascading effects of land-use intensification on multifunctionality.

Methods

- Line 390: Numerous -> how many exactly?

Revised. Please see Line 430 in the revision.

- In general, how did you define whether lower or higher value was desirable for each service/function?

In general, we defined higher soil nutrients, above- and below-ground primary production, biodiversity, forage nutrients and quantity, and cattle stocking density as desirable from the ecosystem service provision perspective, because they represent greater supplies of nutrient retention, carbon storage, biodiversity maintenance, and agricultural production services, respectively. We defined lower water nutrient concentrations, greenhouse gas fluxes, and invasive species diversity as desirable from the ecosystem service provision perspective, because these indicate better water quality, higher greenhouse gas mitigation and thus climate regulation, and greater invasion resistance services. Such definition and interpretation of these indicators in the context of ecosystem services and multifunctionality was based on the system knowledge, which are overall consistent with the literature (e.g., Felipe-Lucia et al. 2018; Qiu et

al. 2019; Qiu et al. 2018; Spiegel et al. 2018; Hölting et al. 2019). To further clarify, we have added these definitions in the Methods (please see Line 458-468 in the revision).

- Where is the information about which management practices were involved and applied to the grasslands? Should be in the manuscript text.

Good suggestion. Per our response above, we have added detailed information on management practices in the main text of the manuscript (e.g., Introduction and Method sections), in addition to those in the Supplementary Information.

References used in this Response:

Boughton, E.H., Quintana-Ascencio, P.F., Bohlen, P.J., Jenkins, D.G. and Pickert, R., 2010. Land-use and isolation interact to affect wetland plant assemblages. *Ecography*, 33(3), pp.461-470.

Boughton, E.H., Quintana-Ascencio, P.F. and Bohlen, P.J., 2011. Refuge effects of *Juncus effusus* in grazed, subtropical wetland plant communities. *Plant Ecology*, 212, pp.451-460.

Felipe-Lucia, M.R., Soliveres, S., Penone, C., Manning, P., van der Plas, F., Boch, S., Prati, D., Ammer, C., Schall, P., Gossner, M.M. and Bauhus, J., 2018. Multiple forest attributes underpin the supply of multiple ecosystem services. *Nature communications*, 9(1), p.4839.

Gounand, I., Harvey, E., Little, C.J. and Altermatt, F., 2018. Meta-ecosystems 2.0: rooting the theory into the field. *Trends in Ecology & Evolution*, 33(1), pp.36-46.

Hölting, L., Jacobs, S., Felipe-Lucia, M.R., Maes, J., Norström, A.V., Plieninger, T. and Cord, A.F., 2019. Measuring ecosystem multifunctionality across scales. *Environmental Research Letters*, 14(12), p.124083.

Pandey, V., Kiker, G.A., Campbell, K.L., Williams, M.J. and Coleman, S.W., 2009. GPS monitoring of cattle location near water features in South Florida. *Applied Engineering in Agriculture*, 25(4), pp.551-562.

Pretty, J., Benton, T.G., Bharucha, Z.P., Dicks, L.V., Flora, C.B., Godfray, H.C.J., Goulson, D., Hartley, S., Lampkin, N., Morris, C. and Pierzynski, G., 2018. Global assessment of agricultural system redesign for sustainable intensification. *Nature Sustainability*, 1(8), pp.441-446.

Qiu, J., Carpenter, S.R., Booth, E.G., Motew, M., Zipper, S.C., Kucharik, C.J., Loheide II, S.P. and Turner, M.G., 2018. Understanding relationships among ecosystem services across spatial scales and over time. *Environmental Research Letters*, 13(5), p.054020.

Qiu, J., Zipper, S.C., Motew, M., Booth, E.G., Kucharik, C.J. and Loheide, S.P., 2019. Nonlinear groundwater influence on biophysical indicators of ecosystem services. *Nature Sustainability*, 2(6), pp.475-483.

Simberloff, D., Von Holle, B. 1999. Positive interactions on nonindigenous species: Invasional meltdown? Biological Invasions, 1, pp. 21–32.

Spiegel, S., Bestelmeyer, B.T., Archer, D.W., Augustine, D.J., Boughton, E.H., Boughton, R.K., Cavigelli, M.A., Clark, P.E., Derner, J.D., Duncan, E.W. and Hapeman, C.J., 2018. Evaluating strategies for sustainable intensification of US agriculture through the Long-Term Agroecosystem Research network. Environmental Research Letters, 13(3), p.034031.

Tamburini, G., Aguilera, G. and Öckinger, E., 2022. Grasslands enhance ecosystem service multifunctionality above and below-ground in agricultural landscapes. Journal of Applied Ecology, 59(12), pp.3061-3071.

Reviewers' Comments:

Reviewer #1:

Remarks to the Author:

The authors addressed all comments in an adequate way. The revised manuscript has much improved, so I recommend its acceptance.

Reviewer #2:

Remarks to the Author:

Manuscript NCOMMS-23-11027A

Comments to the Authors (Guo et al. "Agricultural intensification alters multifunctionality of metaecosystems")

I thank the authors for a thorough revision and detailed responses to my comments. The authors have done, in general, an excellent job improving the manuscript, and have nicely addressed most of my comments and concerns, which I genuinely appreciate. The manuscript has improved in clarity, language, background, and interpretation of results, and now incorporates important information concerning the management practices. I especially appreciate the authors highlighting the result that intensive grassland management actions are manifested in spatially connected unmanaged wetland ecosystems – this is a novel result and is now much more visible. The Discussion has considerably improved, I enjoyed reading it. However, there are a few remaining issues/points which should be considered and clarified further.

The result that grassland management had cascading impacts on embedded natural wetlands where management practices were not imposed, could become clearer in the Abstract. Please highlight the fact that wetlands were not managed, this is not visible in the current version of the Abstract at all, and the reader gets an impression that both grasslands and wetlands are managed. In my opinion, this is the most novel finding and should be emphasized more. For example, the first paragraph of Discussion says this really nicely: "...land-use intensification profoundly altered a broad suite of ecosystem services and their relationships in grasslands, with cascading impacts on embedded natural wetlands where management practices were not directly imposed." For example, lines 45-46 is a good place to mention that managements practices were only applied to grasslands, not on wetlands. The last sentence would be a place to highlight what is said in the last sentence of the first paragraph of the Discussion.

In general, although I appreciate the authors' answer to my question about the trivial nature of the finding that removal of natives and addition of exotics led to increased exotics, this issue is important and deserves some further thinking. The authors explained that several exotic species in intensively managed grasslands did not originate from intentional planting. I agree with the authors that this is an important point and suggest that the authors go further and quantitatively test the role of those exotic species that were not intentionally added, by running the analysis with and without the intentionally added species. If the results remain qualitatively similar with or without intentionally added exotics, it will increase the value of this finding considerably.

I am still slightly confused about ecosystem functions and services in this manuscript. It seems that the authors want to focus on services only, so they should also name their multifunctionality index as "ecosystem service multifunctionality", as defined by Manning et al. 2018 Nat Eco Evo. This should show throughout the manuscript, also in the Abstract. Further, it is important that the authors clearly define what they consider as a "service" as a lot of people working with ecosystem functions (process rates) would call many of the services in this manuscript as functions and pools. Please define this in the Introduction, it does not need to be a long description.

In general, I appreciate the addition of a definition of desired ecosystem service directions to the Methods. Of these, considering high nutrient concentrations as a desired quality, even from ecosystem service point of view, seems to some extent debatable. Greater soil nutrient availability is often related to lower nutrient uptake by plants and to a greater possibility for leaching, both

not so desired qualities, and it is known to show a strong negative correlation with biodiversity. Can you support your choice with some references or explanation?

Detailed comments:

- Lines 79-80: This would read better: "Despite their social-ecological importance, grasslands remain largely ignored in sustainable development...". The deleted part is repetition from previous paragraphs.

- Lines 124-126: This text implies that SN and IM were originally different, i.e., SN grasslands were originally wet prairies while IM grasslands were originally dry grasslands. Is this the case and can it have an impact on how management effects were manifested? Please clarify.

- Line 179: *Most* of the management practices – does this mean that some were applied? Which ones? Discussion (line 230) implies that none were applied but in your response letter you state that the wetlands were grazed. Please clarify.

- Line 228: "...ecosystem functions and services..."

- Line 244: There should be a connecting sentence between the sentence ending "...livestock operations" and the sentence starting "Although P fertilization...", linking the contents of these two sentences together. Now they seem to be totally unrelated.

- In general, this whole second paragraph (lines 235-247) is somehow unconnected to the rest of the story and does not seem to include any interesting results – it feels plain, not exciting and lacks any general conclusions. Can you clarify what is the main result here and what are its consequences?

- Line 270: What is BIR? Never mentioned before, please clarify (not enough to explain it in Methods which come after).

- Lines 256-257: Grazing-resistant instead of grazing-resistance.

- Line 325: In my pdf-file version there is something that does not belong here.

- Line 334: Here it says that bahiagrass has higher nutritional quality than a mixture of different species but on line 242 it says that bahiagrass is low in its nutritional quality. I understand that this can be true but it means that a mixture of species in your system has extremely poor quality which is interesting and must be because most species are grasses?

- Line 394 and elsewhere: Ecosystem service multifunctionality instead of ecosystem multifunctionality. These can be very different things, also in terms of what is considered as desired.

Reviewer #1 (Remarks to the Author):

The authors addressed all comments in an adequate way. The revised manuscript has much improved, so I recommend its acceptance.

We much appreciate your positive feedback on our revision, as well as your previous comments that have greatly helped to improve this paper.

Reviewer #2 (Remarks to the Author):

Manuscript NCOMMS-23-11027A

Comments to the Authors (Guo et al. "Agricultural intensification alters multifunctionality of metaecosystems")

I thank the authors for a thorough revision and detailed responses to my comments. The authors have done, in general, an excellent job improving the manuscript, and have nicely addressed most of my comments and concerns, which I genuinely appreciate. The manuscript has improved in clarity, language, background, and interpretation of results, and now incorporates important information concerning the management practices. I especially appreciate the authors highlighting the result that intensive grassland management actions are manifested in spatially connected unmanaged wetland ecosystems – this is a novel result and is now much more visible. The Discussion has considerably improved, I enjoyed reading it. However, there are a few remaining issues/points which should be considered and clarified further.

Thank you very much for your positive feedback on our revision. Your earlier comments have been really insightful and instrumental for us to further improve and strengthen this work.

The result that grassland management had cascading impacts on embedded natural wetlands where management practices were not imposed, could become clearer in the Abstract. Please highlight the fact that wetlands were not managed, this is not visible in the current version of the Abstract at all, and the reader gets an impression that both grasslands and wetlands are managed. In my opinion, this is the most novel finding and should be emphasized more. For example, the first paragraph of Discussion says this really nicely: "...land-use intensification profoundly altered a broad suite of ecosystem services and their relationships in grasslands, with cascading impacts on embedded natural wetlands where management practices were not directly imposed." For example, lines 45-46 is a good place to mention that managements practices were only applied to grasslands, not on wetlands. The last sentence would be a place to highlight what is said in the last sentence of the first paragraph of the Discussion.

This is a great suggestion. In our revision, we have further emphasized this point in the Lines 46-47 of the Abstract in the revised manuscript. In addition, per your suggestions, we further revised the last sentence of the Abstract by highlighting what has been said in the last sentence of the first paragraph of the Discussion (please see Lines 53-56 in the revision).

In general, although I appreciate the authors' answer to my question about the trivial nature of the finding that removal of natives and addition of exotics led to increased exotics, this issue is important and deserves some further thinking. The authors explained that several exotic species in intensively managed grasslands did not originate from intentional planting. I agree with the authors that this is an important point and suggest that the authors go further and quantitatively test the role of those exotic species that were not intentionally added, by running the analysis with and without the intentionally added species. If the results remain qualitatively similar with or without intentionally added exotics, it will increase the value of this finding considerably.

We highly appreciate this insightful comment, which is very helpful for us to think through more. We agree that conducting additional quantitative assessments to examine the impact of land-use intensification on both intentionally introduced non-native plants and non-introduced non-native plants is crucial and holds significance in enhancing our understanding of pasture invasion resistance. Thus, we have further performed such analyses and included the results in the Supplementary (i.e., Fig. S2, also shown below). We found that our results are overall consistent and robust, where intensively-managed pastures indeed showed reduced invasion resistance capacity (i.e., increased non-native plant richness), regardless of whether including or excluding those plants that are intentionally planted. We further added related Results and Discussion in revised manuscript (please see Lines 177-179, and 256-259 in the revision).

Figure S2. Standardized effect sizes (Hedge's d) of land-use intensification on total non-native plant richness, planted non-native plant richness, and non-planted non-native plant richness. Effect sizes of intensification were estimated by comparing Intensively managed (IM) vs. Semi-natural (SN) grasslands, with error bars representing 95% confidence intervals. Positive Hedge's d denotes a higher indicator value for IM than SN grasslands. Black bars represent significant differences ($\alpha \leq 0.05$) between IM and SN grasslands. Numbers in parentheses mean the sample size for estimating the effect size of each indicator.

I am still slightly confused about ecosystem functions and services in this manuscript. It seems that the authors want to focus on services only, so they should also name their multifunctionality index as "ecosystem service multifunctionality", as defined by Manning et al. 2018 Nat Eco Evo. This should show throughout the manuscript, also in the Abstract. Further, it is important that the authors clearly define what they consider as a "service" as a lot of people working with ecosystem functions (process rates)

would call many of the services in this manuscript as functions and pools. Please define this in the Introduction, it does not need to be a long description.

Thanks for this comment and suggestion. In our revision, we have now changed the “ecosystem multifunctionality” into “ecosystem service multifunctionality” throughout the manuscript (please see examples of Lines 41, 147, 198, 401, 405, 411-412, 463-464, and 480-481). In addition, in our method and supplementary materials, we have clarified our justification of these indicators for ecosystem services based on the literature and their relevance to service provision. To further clarify, we also provided additional justification of our selection of indicators of ecosystem services in the Introduction in Lines 154-156 in the revised manuscript.

In general, I appreciate the addition of a definition of desired ecosystem service directions to the Methods. Of these, considering high nutrient concentrations as a desired quality, even from ecosystem service point of view, seems to some extent debatable. Greater soil nutrient availability is often related to lower nutrient uptake by plants and to a greater possibility for leaching, both not so desired qualities, and it is known to show a strong negative correlation with biodiversity. Can you support your choice with some references or explanation?

This is a good point, and we fully agree with the reviewer that the extent to which an indicator is positive or negative from the ecosystem service perspective can be highly context-dependent. Given that our system is highly nutrient deficient, greater soil nutrient availability in most cases can be beneficial (e.g., for promoting plant growth, and improving soil quality). In addition, our studied grasslands and wetlands constantly receive nutrients from fertilization. Thus, the high soil nutrient levels do not necessarily indicate the lower nutrient uptake by plant or greater possibilities for leaching, but also could be the result of the higher external nutrient input. To further clarify, we have added additional references (Rillig et al., 2023; Le Provost et al., 2023) that justify and support using soil nutrient pools to represent the fertility and nutrient retention levels of ecosystems in Line 471 of the revised manuscript.

References

1. Rillig, M. C. *et al.* Increasing the number of stressors reduces soil ecosystem services worldwide. *Nat. Clim. Chang.* **13**, 478–483 (2023).
2. Le Provost, G. *et al.* The supply of multiple ecosystem services requires biodiversity across spatial scales. *Nat Ecol Evol* **7**, 236–249 (2023).

Detailed comments:

- Lines 79-80: This would read better: “Despite their social-ecological importance, grasslands remain largely ignored in sustainable development...”. The deleted part is repetition from previous paragraphs.

Done. Please see Lines 82-83 in the revision.

- Lines 124-126: This text implies that SN and IM were originally different, i.e., SN grasslands were originally wet prairies while IM grasslands were originally dry grasslands. Is this the case and can it have an impact on how management effects were manifested? Please clarify.

Differences between wet and dry prairies where SN and IM pastures were developed were primarily driven by groundwater levels that are related to geographic locations (e.g., elevation). However, these pasture systems have been substantially modified for agricultural production since 1940s, including heavy drainage and ditching and vegetation clear and conversions. Hence, the past land-use legacy effects from wet and dry prairies are likely to be minimal (as compared to current practices), and most of the observed differences could be attributable to contemporary and ongoing management practices – i.e., including fertilization, higher grazing pressure, conversion of native grasses into productive forages, and more intense drainage.

However, to account for such potential confounding factors, in our Method, we did clarify that “to eliminate the influence of confounding factors such as geographic gradient between SN and IM systems in physiochemical properties, we also included elevation of sampling locations as a covariate”. Please see Lines 519-521 in the revised manuscript.

- Line 179: *Most* of the management practices – does this mean that some were applied? Which ones? Discussion (line 230) implies that none were applied but in your response letter you state that the wetlands were grazed. Please clarify.

Thank you for pointing this out. We apologize for the confusion and want to clarify that most of the management practices were not directly imposed on wetlands, except for the grazing activity. It is because (1) all the natural wetlands embedded within the pastures were not fenced; and (2) therefore, the cattle, when grazing in the pastures, sometimes could also wander into the embedded natural wetlands (e.g., for cooling and additional food sources). However, such grazing pressure would be much lower as compared to the grasslands, and all the rest of the other management practices were only applied to grasslands and not directly imposed on the wetlands. To revise and avoid confusions, we have clarified this in both the Abstract (please see Line 47) and Discussion (see Lines 237 in the revision).

- Line 228: “...ecosystem functions and services...”

Revised. Please Lines 235 in the revision.

- Line 244: There should be a connecting sentence between the sentence ending “...livestock operations” and the sentence starting “Although P fertilization...”, linking the contents of these two sentences together. Now they seem to be totally unrelated.

Sorry for the confusion. Here in this paragraph, we explained effects of land-use intensification on three ecosystem services – soil nutrients, forage quality and quantity, and water quality. These two sentences were referring to the explanations for forage quality and quantity, and water quality services, respectively. However, to enhance clarity, we have made revisions here to improve the flow of discussion (please Lines 251 in the revision).

- In general, this whole second paragraph (lines 235-247) is somehow unconnected to the rest of the story and does not seem to include any interesting results – it feels plain, not exciting and lacks any general conclusions. Can you clarify what is the main result here and what are its consequences?

Thanks for pointing this out. This paragraph is mostly to explain the effects of land-use intensification on three ecosystem services - soil nutrients, forage quality and quantity, and water quality. Per this and previous comment, we have now revised and simplified this paragraph (please see Lines 243-253 in the revised manuscript).

- Line 270: What is BIR? Never mentioned before, please clarify (not enough to explain it in Methods which come after).

Revised (see Lines 278).

- Lines 256-257: Grazing-resistant instead of grazing-resistance.

Revised (see Lines 265)

- Line 325: In my pdf-file version there is something that does not belong here.

Thanks for noting this. No glitch shows up in the word file. We will work with editors to prevent it in the final converted PDF version.

- Line 334: Here it says that bahiagrass has higher nutritional quality than a mixture of different species but on line 242 it says that bahiagrass is low in its nutritional quality. I understand that this can be true but it means that a mixture of species in your system has extremely poor quality which is interesting and must be because most species are grasses?

Typically, bahiagrass has lower nutritional values than normal C₃ grasses (Myer et al., 2011), as shown by the literature. However, when fertilizers are applied, bahiagrass can largely improve its nutrient uptake and retention capacities than other common grasses in Florida (Arthington et al., 2005), resulting in improved nutritional quality.

References

1. Myer, R., Blount, A., Coleman, S. & Carter, J. Forage Nutritional Quality Evaluation of Bahiagrass Selections during Autumn in Florida. *Communications in Soil Science and Plant Analysis* **42**, 167–172 (2011).
2. Arthington, J. D. & Brown, W. F. Estimation of feeding value of four tropical forage species at two stages of maturity^{1,2}. *Journal of Animal Science* **83**, 1726–1731 (2005).

- Line 394 and elsewhere: Ecosystem service multifunctionality instead of ecosystem multifunctionality. These can be very different things, also in terms of what is considered as desired.

This is a great suggestions. We have revised and updated throughout the manuscript (please see our response above).